# Controlling LLM Reasoning Behavior via Guided Decoding

## Abstract

Large Language Models (LLMs) are increasingly being used to simulate human behavior in applications such as educational technology, user modeling, and human–AI interaction. However, LLMs often fail to modulate their reasoning behavior across different intended proficiency levels. Instead, they tend to generate similar reasoning patterns even when explicitly prompted to emulate users with different reasoning behaviors. In this work, we identify this phenomenon as the *Reasoning Prior Bias*, where generation is dominated by a strong prior over certain reasoning patterns. To address this issue, we propose **Guided Decoding**, a decoding-time method for controllably modulating the reasoning behavior of LLMs along a proficiency axis during inference. Our approach fuses token-level logits from two sources: a reference prompt that specifies the model's primary reasoning behavior, and one or more guidance prompts that introduce alternative reasoning tendencies. By adjusting the contribution of these signals during decoding, Guided Decoding enables fine-grained control over the mixture of reasoning behaviors in generated responses. Experiments on multiple question answering benchmarks demonstrate that our method not only modulates task accuracy but also systematically alters the reasoning process. We frame our contribution as controllable reasoning-behavior modulation; establishing fidelity to real human reasoning distributions is a distinct problem that we leave to future work.

## 1 Introduction

Large language models (LLMs) (Jiang et al., 2023a; Touvron et al., 2023; Achiam et al., 2023; Bai et al., 2023) have demonstrated remarkable capabilities across various role-conditioned tasks, such as student simulation (Zhang et al., 2024; Xu et al., 2024; Liu et al., 2024; Schmucker et al., 2024) and user simulation (Davidson et al., 2023; Zhang et al., 2025; Shao et al., 2023; Wu et al., 2025b). In these tasks, LLMs often produce highly capable and structured reasoning processes regardless of the role they are instructed to simulate. However, this tendency presents a limitation: LLMs frequently fail to exhibit the diversity of reasoning behaviors observed in real-world individuals with different levels of expertise (Benedetto et al., 2024; Yuan et al., 2026; Scarlatos et al., 2026; Acquaye et al., 2026). For example, when prompted to answer a question as an "average student", models often generate complete and correct solutions, rather than exhibiting the mistakes, hesitations, or misconceptions that characterize real-world learners.

We refer to this phenomenon as **Reasoning Prior Bias**, where generation is dominated by a strong prior over certain reasoning patterns learned during pretraining and instruction tuning (Qi et al., 2025). As a result, LLMs tend to default to similar reasoning behaviors even when prompted to emulate users with different reasoning characteristics (Zengaffinen et al., 2026). This lack of controllability limits the usefulness of LLMs in scenarios requiring controllable reasoning-behavior simulation, such as educational assessment (Benedetto et al., 2024; Lu & Wang, 2024), personalized tutoring (Gao et al., 2025; Tang et al., 2025), psychological measurement (Yang et al., 2024) and healthcare (Li et al., 2024a). Although recent work such as role-playing via in-context learning (Kong et al., 2024; Gao et al., 2025; Xu et al., 2024; Wang et al., 2025a) or post-training (Shao et al., 2023; Yu et al., 2024; Wang et al., 2024a) attempts to control LLM behavior, these approaches often require detailed persona descriptions or extensive interaction histories, which are impractical to scale across large populations. Broader demographic role-play improves scalability but lacks the resolution to distinguish subtle differences in reasoning behavior within the same role. Consequently, existing methods

struggle to support **controllable reasoning behavior simulation**, particularly for modulating reasoning behavior toward users with average or limited reasoning proficiency.

To address this challenge, we observe that while prompting is a flexible tool for steering LLM behavior, it often lacks the precision to reliably alter the reasoning trajectory during generation. Recent work suggests that decoding-time interventions can influence reasoning dynamics (Li et al., 2022; Wang et al., 2025b), for example by modifying token probabilities to encourage exploration or mimic underthinking. These observations raise an important research question: *can we simulate individuals with different reasoning behaviors not only by changing the instruction prompts, but also by shaping how the model reasons during decoding?*

To this end, we propose *Guided Decoding*, a decoding-time method for dynamically controlling the reasoning behavior of LLMs during inference. Specifically, our method fuses token-level logits from two sources: (1) a reference prompt that specifies the model's primary reasoning behavior, and (2) one or more guidance prompts that introduce alternative reasoning tendencies. By interpolating between these signals at each decoding step, Guided Decoding enables fine-grained control over the mixture of reasoning behaviors. Importantly, Guided Decoding does not require parameter updates or model fine-tuning; it uses fixed reference and guidance prompts constructed before evaluation, making it lightweight and broadly applicable.

Our contributions are summarized as follows: (1) We identify the **Reasoning Prior Bias** in LLMs, where generation is dominated by strong priors over certain reasoning patterns, limiting behavioral diversity. (2) We propose **Guided Decoding**, a decoding-time approach for controllably steering reasoning behavior in LLMs. (3) Extensive experiments demonstrate that Guided Decoding can systematically alter reasoning processes and modulate task performance across a spectrum of reasoning behaviors.

## 2 Related Work

**Role-Playing with Language Models.**   Role-playing is a paradigm for aligning language models with specific behaviors, tasks, or user expectations. Prior work (Chen et al., 2024; Yuan et al., 2026) has established several types of role-playing settings. Demographic role-playing (Jiang et al., 2023b; Hong et al., 2024; Kong et al., 2024; LI et al., 2024; Wu et al., 2025a; Li et al., 2026) focuses on simulating a demographic of people, such as age, gender, and occupation. Character role-playing (Shao et al., 2023; Li et al., 2024b; Shaikh et al., 2025), on the other hand, simulates specific individuals, such as fictional or historical personas and preset individuals. From a technical standpoint, role-playing is typically implemented via in-context learning, using instructions to change LLMs' behavior (Wang et al., 2024a; Dai et al., 2025; Scarlatos et al., 2026). Some works incorporate agent architectures or memory modules (Gao et al., 2025; Xu et al., 2024; Hong et al., 2024; Wang et al., 2024b) to sustain consistency in roles over long interactions. Others adopt parametric approaches, fine-tuning the model with domain-specific data to internalize persona characteristics (Shao et al., 2023; Yu et al., 2024; Wang et al., 2024a). While these methods effectively control style or behavior, they generally assume static or consistent personas and do not aim to control the underlying reasoning ability or proficiency level. Instead of assigning a character identity, we aim to simulate individuals with varying cognitive abilities. Technically, our method differs from prior work by controlling proficiency at decoding level, rather than prompt engineering or model fine-tuning. This enables fine-grained, dynamical, and model-agnostic control over reasoning behavior.

**LLM Decoding.**   Decoding plays a key role in shaping the output behavior of LLMs. Some basic decoding methods (e.g., greedy (Germann, 2003) and beam search (Freitag & Al-Onaizan, 2017)) select tokens with the highest logits at each step. These methods perform well in many goal-driven tasks but often suffer from repetitive outputs. Additionally, there is growing interest in decoding methods that improve not only the fluency but also reasoning capability of LLM. *Contrastive decoding* (Li et al., 2022; O'Brien & Lewis, 2023; Phan et al., 2024; Chia et al., 2023; Liu et al., 2021) improves that by contrasting the outputs of a strong expert model with a weak amateur model. In parallel, decoding-time penalty mechanisms (Su & Collier, 2023; Wang et al., 2025b; Lightman et al., 2024; Heineman et al., 2024) have been proposed to alter the generation process. Repetition penalties (Su & Collier, 2023) discourage redundant token sequences, while Thought Switching Penalty (TIP) (Wang et al., 2025b) are designed to improve reasoning depth by lowering the logits of *thought-switching* tokens, encouraging deeper thinking. Steering methods (Dong et al., 2026;

Table 1: An example of GPT-4o responses under different reasoning behavior prompts. The accuracy refers to the performance on the entire GSM8K dataset. The full prompts are shown in Appendix D.

| You are an {**Role**} student. Please answer the question: Weng earns \$12 an hour for babysitting. Yesterday, she just did 50 minutes of babysitting. How much did she earn? | | |
|---|---|---|
| **Role** | **Model Response** | **Acc** |
| **A** | To solve this, we first convert 50 minutes into hours: $\frac{50}{60} = \frac{5}{6}$. Then, $\frac{5}{6} \times 12 = 10$. Answer: Weng earned **\$10.00** for 50 minutes of babysitting. | 97.1 |
| **B** | Weng babysat for 50 minutes. That's $50 \div 60 = 0.833$ hours. $12 \times 0.833 \approx 10$. So she earned about **\$10**. | 96.6 |
| **C** | Weng earns \$12 an hour. She worked for 50 minutes, which is close to an hour. Maybe she earned around \$10. I'll just say **\$10**. | 95.7 |
| **D** | Weng earns \$12 an hour. She worked 50 minutes, so maybe like \$10? I think she earned **\$10**. | 94.3 |

Zhuang et al., 2026; Kortukov et al., 2026) aim to steer the model activations to achieve different model behavior, like refusal. However, to our best knowledge, no prior work explicitly targets the controllable proficiency simulation problem through decoding. While existing methods are non-Monotonic and lack of fine-grained and probabilistic control, our work fills this gap by proposing a decoding-time strategy that enables fine-grained control over LLM reasoning ability.

Our method is related to logit-fusion approaches such as contrastive decoding (Li et al., 2022; O'Brien & Lewis, 2023) and DExperts (Liu et al., 2021), but differs in objective and construction. These methods contrast an expert stream with an amateur/anti-expert stream to *improve* generation quality or suppress an undesirable attribute, typically using separate models and a binary, unidirectional intervention. A direct off-the-shelf comparison is not well-defined for our proficiency-control setting because there are no separately trained expert/anti-expert proficiency models; replacing those models with our prompt-conditioned reference/guidance streams reduces the comparison to prompt-stream logit interpolation. In contrast, Guided Decoding (i) uses a single model with prompt-induced streams rather than separately trained experts; (ii) intentionally *preserves* both reference and error-pattern guidance streams instead of suppressing the weaker one; (iii) provides graded, bidirectional control through a continuous level parameter $\lambda_l$; and (iv) aggregates multiple guidance streams via similarity-aware weighting. We include direct good-vs-poor logit interpolation and uniform multi-guidance fusion as decoding-time baselines (Section 5.4) to isolate the effect of these design choices.

## 3 Preliminary

### 3.1 Observation of Reasoning Prior Bias

We begin by empirically examining whether role-conditioned prompting induces meaningful variation in reasoning behavior. However, we observe that LLMs tend to produce highly similar reasoning patterns regardless of the role-conditioned instructions. We refer to this phenomenon as *Reasoning Prior Bias*, where generation is dominated by strong priors over certain reasoning patterns learned during pretraining and instruction tuning. Table 1 illustrates this phenomenon using GPT-4o prompted to simulate students with different reasoning characteristics, with academic role descriptors ranging from A-score to D-score, associated with students of varying proficiency levels. While surface tone and confidence vary across roles, the underlying reasoning trajectory and numerical outcome remain largely unchanged. Correspondingly, the response accuracy shows minimal variation across different role prompts.

This observation is not an isolated example, but reflects a systematic pattern validated across our broader experiments. As shown in Figure 3 and Table 2, prompting-based methods consistently maintain similar performance across intended roles, with limited controllable variation and irregular transitions between target reasoning behaviors. We attribute this phenomenon to the model's reasoning prior (Lin et al., 2024; Favero

et al., 2024; Zengaffinen et al., 2026; Acquaye et al., 2026; Yuan et al., 2026), acquired during pretraining and instruction tuning, which biases generation toward certain structured reasoning patterns. Consequently, role conditioning through prompting alone is often insufficient to override the dominant reasoning tendencies encoded in the model.

### 3.2 Problem Statement

Let $\mathcal{P}$ denote the space of prompts, and $\mathcal{Y}$ denote the space of possible outputs. To make reasoning behavior measurable, we introduce a discrete set of proficiency levels $\mathcal{L} = \{0, 1, \ldots, N\}$ with $N = 10$, yielding 11 settings, where $L \in \mathcal{L}$ represents the resulting reasoning proficiency of the generated response. The two endpoints $L = 0$ and $L = N$ are boundary conditions, and the 9 intermediate levels $L \in \{1, \ldots, 9\}$ are the ones we report. These levels are defined based on observable task performance, and serve as a practical proxy for the quality of the reasoning process.

Given a prompt $P \in \mathcal{P}$ and a target proficiency level $L \in \mathcal{L}$, the generation process produces an output $O = f(L, P) \in \mathcal{Y}$, where $f(\cdot)$ denotes the behavior-controlled generation function. Intuitively, different levels correspond to different reasoning tendencies during generation, which lead to different performance outcomes.

In practice, when reasoning behavior improves gradually, we expect the resulting task performance to change smoothly as well. Therefore, we evaluate controllability by examining whether performance varies consistently across the specified levels. To capture this property, we aim to construct $f$ such that the following conditions are satisfied:

**Property 1: Local Consistency (Slope Deviation Score, SDS).** To ensure consistent transitions between adjacent levels of simulated reasoning proficiency, we define the empirical slope at level $L$ as:

$$\Delta_L = \mathbb{E}_{P \sim \mathcal{P}}[\mathcal{Q}(f(L, P)) - \mathcal{Q}(f(L-1, P))], \tag{1}$$

where $\mathcal{Q}(\cdot)$ denotes task-specific evaluation metrics (e.g., accuracy, correctness, or reasoning quality). Let the ideal linear slope be defined as:

$$\alpha = \frac{\mathbb{E}_{P \sim \mathcal{P}}[\mathcal{Q}(f(N, P))] - \mathbb{E}_{P \sim \mathcal{P}}[\mathcal{Q}(f(0, P))]}{N} \tag{2}$$

This ideal slope $\alpha$ represents the expected average performance gain per proficiency level under a perfectly linear progression from the lowest to the highest proficiency. The *Slope Deviation Score (SDS)* then measures the total deviation from this ideal slope across all transitions:

$$\text{SDS} = \sum_{L=1}^{N} |\Delta_L - \alpha| \tag{3}$$

A low SDS implies that the model's performance increases (or decreases) consistently across levels, preserving a smooth and predictable progression.

**Property 2: Global Alignment (Proficiency Deviation Score, PDS).** Let the ideal performance at level $L$ be defined as a linear interpolation between the endpoints:

$$\hat{\mathcal{Q}}(L) = \alpha \cdot L + \mathcal{Q}(f(0, P)), \tag{4}$$

We define the *Proficiency Deviation Score (PDS)* as the total deviation between actual and ideal performance across all intermediate levels:

$$\text{PDS} = \sum_{L=1}^{N-1} \left| \mathbb{E}_{P \sim \mathcal{P}}[\mathcal{Q}(f(L, P))] - \hat{\mathcal{Q}}(L) \right| \tag{5}$$

A lower PDS indicates that the model follows the expected proficiency–performance trajectory, avoiding reversals or irregular deviations across levels.

**Property 3: Behavior Transitions.** The transition between outputs $f(L_1, P)$ and $f(L_2, P)$ for $L_1 < L_2$ should correspond to reasoning transitions, such as fewer logical errors and increased accuracy, rather than random or stylistic variation.

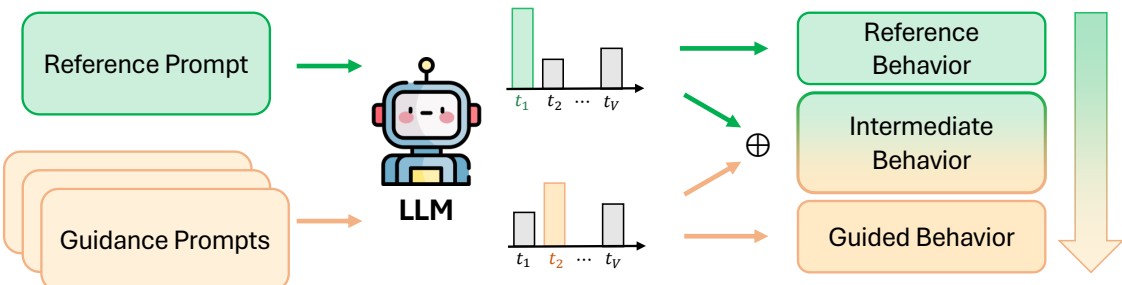

Figure 1: Overview of the Guided Decoding framework for controlling reasoning behavior in LLMs. A reference prompt induces the model's primary reasoning trajectory, while guidance prompts introduce alternative reasoning tendencies. During decoding, token logits from the reference stream are combined with those from the guidance streams through a fusion mechanism, enabling controllable modulation of the generated reasoning process.

## 4 Method

As shown in Figure 1, to address the reasoning prior bias, we propose **Guided Decoding**, a decoding-time strategy that integrates two reasoning streams during generation: a primary reasoning stream induced by a reference prompt and auxiliary reasoning streams induced by guidance prompts. These streams are combined at the token level through a logit fusion mechanism, enabling controllable modulation of the reasoning process. The overall framework consists of two components: prompt construction and guided decoding.

### 4.1 Prompts Construction

To induce distinct reasoning signals during generation, we construct two types of prompts: a reference prompt and a set of guidance prompts. These prompts are designed to produce different reasoning trajectories, which are later combined during decoding. An example is illustrated in Figure 2, where the reference prompt produces a standard reasoning trajectory, while the guidance prompts introduce alternative reasoning patterns. More details can be found in Appendix D.

**Reference Prompt.** The reference prompt is designed to elicit the model's primary reasoning behavior for the target task. It typically includes explicit instructions and structured examples that encourage coherent and logically consistent reasoning. The reference prompt therefore produces a reasoning trajectory that reflects the model's default reasoning tendency under standard prompting conditions.

**Guidance Prompts.** Guidance prompts are introduced to provide reasoning signals that differ from the primary reasoning trajectory induced by the reference prompt. By incorporating these prompts during decoding, the model can access multiple reasoning tendencies instead of relying solely on the dominant reasoning prior. In practice, guidance prompts can be constructed using representative task examples and asking the language model to produce alternative reasoning patterns and corresponding responses. Depending on the application, practitioners may also manually design or select guidance prompts that reflect *task-relevant* reasoning behaviors. These prompts then serve as auxiliary reasoning sources during decoding.

### 4.2 Guided Decoding

Recent studies have shown that decoding strategies can influence the reasoning trajectory of large language models during generation (Wang et al., 2025b; O'Brien & Lewis, 2023). Motivated by these findings, we propose *Guided Decoding*, a decoding-time framework that modulates the model's reasoning behavior by combining logits produced under multiple prompts.

**Standard Decoding.** In original autoregressive decoding, a language model generates text by predicting one token at a time. At each position $p$, the model outputs a logit vector $\mathbf{z}_p \in \mathbb{R}^{|\mathcal{T}|}$, where $|\mathcal{T}|$ is the vocabulary

Figure 2: An illustration of Prompts Construction

size. The probability of generating a token $v \in \mathcal{T}$ is computed via the softmax function:

$$P(x_p = v \mid x_{<p}) = \frac{\exp(\mathbf{z}_{p,v})}{\sum_{v' \in \mathcal{T}} \exp(\mathbf{z}_{p,v'})} \tag{6}$$

where $\mathbf{z}_{p,v}$ is the token logit for token v. This decoding process relies entirely on a single prompt and the model's default reasoning trajectory. As a result, it inherently reflects the most likely or confident output of the model, which often corresponds to high-proficiency, expert-like reasoning—regardless of the role or ability level the user intends to simulate.

Under this formulation, decoding is conditioned on a single prompt and follows the reasoning trajectory implicitly induced by that prompt. Consequently, generation tends to follow the dominant reasoning patterns encoded in the model, which limits the controllability of reasoning behavior once the prompt is fixed.

**Guided Decoding via Logit Fusion.** To enable controllable modulation of reasoning behavior, we propose a Guided Decoding framework with an auxiliary control mechanism that modulates the token-level logits using multiple prompts. ***Reference Logits*** ($\mathbf{z}_p^{\text{ref}}$). The reference logits are the *raw (pre-softmax) token-logit vector* produced by prompting the LLM with a reference prompt $P_{\text{ref}}$ designed to elicit the model's primary reasoning behavior for the target task. To mitigate the dominance of the model's default reasoning prior, we apply a down-weighting factor to this signal: $\tilde{\mathbf{z}}_p^{\text{ref}} = \beta \cdot \mathbf{z}_p^{\text{ref}}$, where $\mathbf{z}_p^{\text{ref}} \in \mathbb{R}^{|\mathcal{T}|}$ is the raw reference logit vector at position $p$, and $\beta \in (0, 1]$ is a hyperparameter that moderates the influence of the reference reasoning stream, allowing balanced integration with auxiliary reasoning signals from guidance prompts. We emphasize that the reference, guidance, aggregated guidance, and final fused streams are all represented as raw logits. Softmax probabilities are used only to compute scalar similarity weights for guidance aggregation; the actual stream aggregation and final decoding remain in raw logit space.

***Guidance Logits*** ($\mathbf{z}_p^{\text{g}}$). The guidance logits are computed from a set of $K$ guidance prompts $\{P_{\text{g}}^k\}_{k=1}^K$, each designed to induce alternative reasoning tendencies for the task. For each prompt $P_{\text{g}}^k$, the model produces token logits $\mathbf{z}_p^k$ at position $p$.

To encourage diverse reasoning signals, we adopt a similarity-aware aggregation strategy that weights each guidance signal according to its divergence from the reference reasoning stream. Specifically, we first convert the scaled reference logits and each guidance logit vector into softmax probability distributions:

$$\mathbf{p}_p^{\text{ref}} = \text{softmax}(\tilde{\mathbf{z}}_p^{\text{ref}}), \qquad \mathbf{p}_p^k = \text{softmax}(\mathbf{z}_p^k).$$

We then compute a distance-like score between each guidance distribution and the reference distribution using the complement of cosine similarity: $s^k = 1 - \cos\left(\mathbf{p}_p^k, \mathbf{p}_p^{\text{ref}}\right)$. A larger value of $s^k$ indicates greater deviation from the reference reasoning trajectory. The probability distributions are used only for computing the scalar weights $\alpha^k$; after these weights are obtained, guidance aggregation is performed over the raw guidance logits.

---

**Algorithm 1** Guided Decoding (one generation)

---

**Require:** reference prompt $P_{\text{ref}}$, guidance prompts $\{P_{\text{g}}^k\}_{k=1}^K$, level $l \in \{0, \ldots, 10\}$, reference scale $\beta$
1: $\lambda \leftarrow l/10;$     shared prefix $y_{<p} \leftarrow \varnothing$
2: **while** not finished **do**
3:      compute raw logits $\mathbf{z}_p^{\text{ref}}$ from $[P_{\text{ref}}; y_{<p}]$
4:      compute raw logits $\mathbf{z}_p^k$ from each $[P_{\text{g}}^k; y_{<p}]$
5:      $\tilde{\mathbf{z}}_p^{\text{ref}} \leftarrow \beta\, \mathbf{z}_p^{\text{ref}}$
6:      $\mathbf{p}_p^{\text{ref}} \leftarrow \text{softmax}(\tilde{\mathbf{z}}_p^{\text{ref}});$     $\mathbf{p}_p^k \leftarrow \text{softmax}(\mathbf{z}_p^k)$ for all $k$
7:      $s^k \leftarrow 1 - \cos(\mathbf{p}_p^k, \mathbf{p}_p^{\text{ref}})$ for all $k$
8:      **if** $\sum_k s^k = 0$ **then**     $\alpha^k \leftarrow 1/K$     **else**     $\alpha^k \leftarrow s^k / \sum_j s^j$
9:      **end if**
10:      $\mathbf{z}_p^{\text{g}} \leftarrow \sum_k \alpha^k\, \mathbf{z}_p^k$
11:      $\mathbf{z}_p \leftarrow \lambda\, \tilde{\mathbf{z}}_p^{\text{ref}} + (1 - \lambda)\, \mathbf{z}_p^{\text{g}}$
12:      decode next token from $\mathbf{z}_p$; append it to shared prefix $y_{<p}$
13: **end while**

---

This avoids computing cosine similarity directly on unnormalized logits while also avoiding probability-logit fusion. The guidance logits are then aggregated as

$$\mathbf{z}_p^{\text{g}} = \sum_{k=1}^K \alpha^k \cdot \mathbf{z}_p^k, \quad \alpha^k = \frac{s^k}{\sum_{j=1}^K s^j}. \tag{7}$$

This similarity-based aggregation allows guidance streams that deviate more strongly from the reference reasoning trajectory to exert greater influence on the fused logits, thereby introducing controlled variation in the reasoning process.

***Logit Fusion.*** The final fused logits $\mathbf{z}_p$ used for token prediction are computed by interpolating between the reference and guidance logits:

$$\mathbf{z}_p = \lambda \cdot \tilde{\mathbf{z}}_p^{\text{ref}} + (1 - \lambda) \cdot \mathbf{z}_p^{\text{g}}. \tag{8}$$

Here $\lambda \in [0, 1]$ controls the relative contribution of the reference and guidance reasoning streams. To relate this parameter to discrete simulation levels, we define $\lambda$ using a level index $l \in \{0, 1, \ldots, N\}$ with $N = 10$ as $\lambda = \frac{l}{N}$, so the scale spans 11 settings indexed $l \in \{0, \ldots, 10\}$: the two endpoints $l = 0$ (pure guidance) and $l = 10$ (pure reference), together with the 9 intermediate levels $l \in \{1, \ldots, 9\}$ that we report. The next token is decoded from the fused logits $\mathbf{z}_p$ using the same decoding rule as the base model. Larger values of $\lambda$ place greater emphasis on the reference reasoning stream, while smaller values increase the influence of guidance signals.

Algorithm 1 summarizes the full decoding loop, making explicit the prefix sharing, probability-space similarity computation, normalization, and the zero-similarity edge case. At each step, the reference and guidance streams share a single generated prefix $y_{<p}$; the selected token is appended to every stream so that all next-step logits are computed under the same continuation. In implementation, the streams are batched into one forward pass, but each stream keeps a separate KV cache because the prompt contexts differ.

## 5 Experiments

### 5.1 Experiment settings

**Datasets.** To evaluate controllable reasoning behavior under different task settings, we adopt benchmarks commonly used in prior work on student and user simulation (Lu & Wang, 2024; Benedetto et al., 2024). We experiment on four datasets. The **RACE** dataset (Lai et al., 2017) is derived from English reading

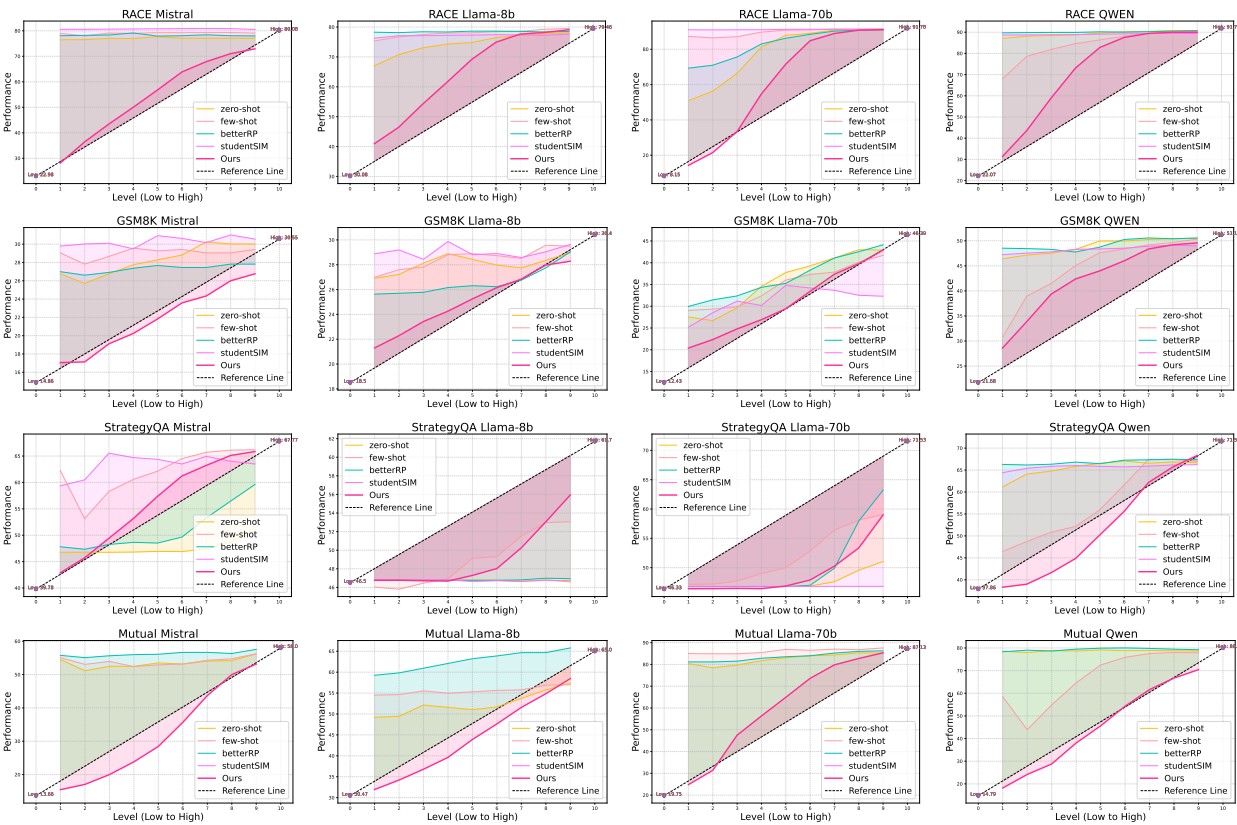

Figure 3: Main results across four datasets and four model backbones. Each curve represents model performance across nine reasoning simulation levels. The reference trajectory is defined as the ideal linear interpolation between its two endpoint settings. The vertical deviation between each method's curve and the reference trajectory corresponds to the Proficiency Deviation Score (PDS), while the deviation in adjacent-level slopes from the reference trajectory reflects the Slope Deviation Score (SDS).

comprehension exams. The **GSM8K** dataset (Cobbe et al., 2021) is a fundamental mathematical question-answering dataset that requires multi-step reasoning for resolution. The **StrategyQA** dataset (Geva et al., 2021) focuses on open-domain commonsense questions. To further evaluate generalization, we include a multi-turn dialogue benchmark. The **Mutual** dataset (Cui et al., 2020) is a multi-turn dialogue benchmark evaluating multi-turn conversations with reasoning-intensive response selection. More details can be found in Appendix A.

**Models.** We evaluate Guided Decoding on a diverse set of open-source large language models from three model families. For the main experiments, we use instruct checkpoints: Mistral-7B-Instruct-v0.3 (Jiang et al., 2023a), LLaMA3-8B-Instruct, LLaMA3-70B-Instruct (Touvron et al., 2023), and Qwen3-Instruct (Yang et al., 2025) as backbone models, with tokenizers following the corresponding instruct checkpoints. These models cover a broad range of parameter scales, allowing us to examine whether the proposed method remains effective across models with different reasoning capacities. To further study scalability, we conduct additional experiments on larger instruct variants from the Mistral and Qwen families, including Mistral-8B, Mistral-24B, Qwen3-8B, Qwen3-14B, and Qwen3-32B. All models are used without parameter updates, and Guided Decoding is applied solely at inference time by combining the token-level outputs induced by the reference and guidance prompts.

**Baselines.** To evaluate the effectiveness of our Guided Decoding framework, we compare against four representative baselines, covering prompt engineering, role-play conditioning, and student simulation approaches. Specifically, we include **zero-shot prompting** (Kojima et al., 2022), and **few-shot prompting** (Wei et al., 2022). These approaches represent standard prompting strategies. **betterRP** (Kong et al., 2024) improves

role-playing by injecting task-aware persona descriptions into the prompt. **studentSIM** (Benedetto et al., 2024) designs a student simulation method by involving the estimation of question difficulty, which helps simulation alignment for different student levels.

**Metrics.** To evaluate controllable reasoning behavior, we use task accuracy together with two metrics introduced in Section 3.2, *Proficiency Deviation Score (PDS)* and *Slope Deviation Score (SDS)*. (1) **Proficiency Deviation Score (PDS)**, as shown in Eq. equation 5, quantifies the global alignment between the actual proficiency–performance curve and the ideal linear performance trajectory across simulated proficiency levels. It penalizes reversals or inconsistent deviations in expected performance, and is particularly sensitive to large gaps or misordered trends across intermediate levels. (2) **Slope Deviation Score (SDS)**, as shown in Eq. equation 3, captures the local consistency of reasoning transitions by measuring how closely the empirical slope between adjacent levels aligns with the ideal uniform slope $\alpha$. It reflects the smoothness and monotonicity of transitions between adjacent simulation levels.

**Implementation Details.** In our experiments, we simulate 11 discrete proficiency settings indexed $l \in \{0, \ldots, 10\}$ with $\lambda = l/10$, comprising two endpoints ($l = 0, 10$) and nine intermediate levels ($l \in \{1, \ldots, 9\}$), corresponding to scores from 0 to 100 in increments of 10 (i.e., $\{0, 10, 20, ..., 100\}$). For each dataset, we construct one fixed reference prompt and a fixed set of guidance prompts at the dataset level, and reuse them across all evaluation questions. Demonstrations are sampled from the training split, while all reported results are evaluated on the corresponding test split; test labels are not used to construct reference or guidance demonstrations. GPT-4o is used offline to generate fixed reference/guidance answers and for diagnostic error analysis, but it is not used as the test-time decoder. We set $\beta$ as 0.5 empirically and use $K = 3$ guidance prompts in the main setting. All main generations are produced using greedy decoding from the fused logits, and final answers are extracted from the required `<Answer>...</Answer>` tag. Experiments are conducted on a combination of NVIDIA RTX A5000 GPUs (24GB) and NVIDIA A100 GPUs (80GB), depending on model size.

### 5.2 Main Results

The results across four datasets and four backbone models are presented in Figure 3 and Table 2. We evaluate each method's ability to produce controllable reasoning behavior across the nine intermediate simulation levels, which correspond to different proficiency bands.

**Global Alignment: Proficiency Deviation Score (PDS).** Across most settings, Guided Decoding achieves the lowest PDS, substantially reducing the deviation from the ideal performance trajectory. In many cases, PDS is reduced by tens to hundreds of points. For example, on RACE with Mistral, our method achieves a PDS of **30.78**, compared to as high as **262.62** from the studentSIM baseline, corresponding to approximately a 7× reduction. This substantial improvement indicates that Guided Decoding more closely follows the intended performance progression across simulation levels.

In some cases, such as StrategyQA with LLaMA models, we observe elevated PDS values across all methods, and Guided Decoding is not uniformly best in every cell. Guided Decoding helps most when the reference and guidance prompts induce meaningfully different reasoning trajectories that can be smoothly interpolated. It helps least when (i) the base model is already unstable on the task, e.g., StrategyQA requires commonsense and implicit world knowledge, so errors often stem from missing or hallucinated facts rather than controllable reasoning-pattern changes; (ii) the answer format is binary (True/False), which limits the granularity of performance modulation compared to multi-choice datasets; or (iii) the guidance prompts do not cover the dataset's dominant failure mode. Thus the key limiting factor is not model size alone but the alignment between the task's dominant error modes and the available guidance patterns. These results suggest that although Guided Decoding improves controllability, the overall effectiveness remains influenced by the underlying model quality and dataset characteristics.

**Local Consistency: Slope Deviation Score (SDS).** In addition to global alignment, Guided Decoding consistently achieves the lowest SDS across most settings, indicating its ability to produce smooth and monotonic performance transitions across adjacent simulation levels. In many cases, our method reduces SDS by more than half compared to competing baselines.

Table 2: Quantitative results across four datasets and four model backbones. Lower PDS/SDS indicates closer alignment with the reference performance trajectory and more consistent performance transitions across simulation levels.

| Dataset | Methods | Mistral | | Llama-8b | | Llama-70b | | Qwen | |
|---|---|---|---|---|---|---|---|---|---|
| | | PDS↓ | SDS↓ | PDS↓ | SDS↓ | PDS↓ | SDS↓ | PDS↓ | SDS↓ |
| RACE | zero-shot | 229.20 | 45.05 | 176.47 | 28.43 | 254.64 | **43.34** | 291.77 | 52.07 |
| | few-shot | 247.54 | 45.55 | 208.46 | 35.32 | 295.82 | 62.66 | 245.05 | 40.71 |
| | betterRP | 240.40 | 45.75 | 212.90 | 39.22 | 295.83 | 45.18 | 298.72 | 54.73 |
| | studentSIM | 262.62 | 45.74 | 201.70 | 38.10 | 370.72 | 66.62 | 289.21 | 55.07 |
| | Ours | **30.78** | **15.24** | **89.44** | **19.92** | **112.65** | 49.79 | **134.49** | **44.26** |
| GSM8K | zero-shot | 50.01 | 9.29 | 33.06 | 7.47 | 57.91 | 14.85 | 112.62 | 19.50 |
| | few-shot | 56.93 | 12.17 | 36.36 | 7.02 | 51.46 | 14.97 | 73.52 | 15.80 |
| | betterRP | 44.07 | 11.72 | 20.33 | 6.31 | 64.70 | 13.00 | 115.71 | 21.55 |
| | studentSIM | 68.35 | 11.79 | 41.29 | 9.26 | 58.42 | 22.51 | 106.89 | 21.84 |
| | Ours | **9.56** | **4.26** | **7.68** | **2.54** | **12.84** | **6.96** | **53.52** | **12.49** |
| StrategyQA | zero-shot | 67.25 | 18.38 | 66.20 | 12.34 | 101.45 | 15.88 | 100.23 | 21.44 |
| | few-shot | 74.97 | 23.16 | 45.67 | 9.89 | **62.84** | **11.00** | **27.70** | 16.87 |
| | betterRP | 38.94 | 13.72 | 65.54 | 11.99 | 78.44 | 21.06 | 110.84 | 25.89 |
| | studentSIM | 89.37 | 22.77 | 66.19 | 12.25 | 109.08 | 20.16 | 103.16 | 25.11 |
| | Ours | **19.99** | **8.11** | **45.38** | **9.69** | 87.28 | 14.96 | 30.14 | **11.70** |
| Mutual | zero-shot | 159.24 | 33.77 | 57.10 | 19.38 | 261.13 | 49.05 | 281.69 | 52.06 |
| | few-shot | 163.31 | 34.34 | 82.13 | 25.13 | 293.99 | 51.31 | 176.96 | 50.73 |
| | betterRP | 183.28 | 33.67 | 134.73 | 21.08 | 270.60 | 48.82 | 287.34 | 51.27 |
| | Ours | **36.97** | **14.68** | **30.63** | **4.63** | **72.45** | **23.92** | **21.90** | **13.44** |

Table 3: Quantitative reasoning quality analysis

| Level (low to high) | 1 | 3 | 5 | 7 | 9 |
|---|---|---|---|---|---|
| Logical errors | 115 | 125 | 109 | 104 | 95 |
| False prior knowledge | 9 | 7 | 5 | 3 | 4 |
| Surface-heuristic mistakes | 72 | 60 | 57 | 47 | 47 |
| **Avg. errors per request** | **1.96** | **1.92** | **1.71** | **1.54** | **1.46** |

For example, on Mutual with LLaMA-8B, Guided Decoding achieves an SDS of **4.63**, while other methods range from 19.38 to 25.13, indicating more than a 4× improvement in stepwise consistency. In contrast, zero-shot and few-shot prompting exhibit frequent fluctuations in SDS due to their reliance on natural language conditioning, which lacks the granularity needed to precisely modulate reasoning dynamics across levels. Similarly, role-play prompting methods fail to enforce stable transitions between levels because high-level persona descriptions do not provide sufficient control over the model's internal generation process.

**Visualization Insights.** The performance curves provide qualitative evidence for the effectiveness of Guided Decoding. Unlike baseline methods, whose trajectories frequently exhibit non-monotonic trends, Guided Decoding produces monotonic non-decreasing trajectories in 14/16 main dataset-backbone settings. The two exceptions are StrategyQA with LLaMA-8B and LLaMA-70B, where the curves contain only small local dips. Specifically, we observe frequent violations of monotonicity in methods such as studentSIM and BetterRP on GSM8K and RACE, where intermediate levels occasionally outperform both lower and higher levels. These reversals indicate unstable reasoning dynamics, suggesting that prompt-based approaches alone cannot reliably regulate the model's generation process. Furthermore, in many cases, model performance remains consistently high across simulation levels. This observation aligns with the reasoning prior bias discussed earlier: LLMs tend to follow dominant reasoning patterns learned during pretraining and instruction tuning,

Table 4: Model responses across different reasoning simulation levels for a math problem.

| **Question:** Vincent can buy flowers in packages of 3 for $2.50 or in packages of 2 for $1. How much money does he save by buying 18 flowers at the better price? |
|---|

| Reasoning | Model Response |
|---|---|
| **Reference** | Vincent can buy flowers in packages of 3 for $2.50 or in packages of 2 for $1. So, if he buys 18 flowers, he can buy 6 packages of 3 flowers for $2.50 each, or 9 packages of 2 flowers for $1 each. The total cost of 6 packages of 3 flowers is $6 \times \$2.50 = \$15.00$. The total cost of 9 packages of 2 flowers is $9 \times \$1 = \$9.00$. So, Vincent saves $15.00 - $9.00 = **$6.00** by buying 18 flowers at the better price. |
| **Level (6)** | Vincent can buy flowers in packages of 3 for $2.50 or in packages of 2 for $1. If he buys 18 flowers, he can buy 6 packages of 3 flowers for $2.50 each, or 9 packages of 2 flowers for $1 each. So, he saves $\$2.50 \times 6 - \$1 \times 9 = $ **$6.00**. |
| **Level (3)** | Vincent can buy flowers in packages of 3 for $2.50 or in packages of 2 for $1. So, if he buys 18 flowers, he will buy 6 packages of 3 flowers and 2 packages of 2 flowers. The total cost is $6 \times 2.50 + 2 \times 1 = $ **$15.00**. |
| **Level (1)** | Vincent can buy flowers in packages of 3 for $2.50 or in packages of 2 for $1. He buys 18 flowers at the better price. He saves $\$2.50 \times 18 = $ **$45.00**. |

making it difficult to naturally produce intermediate reasoning behaviors without explicit control during decoding.

### 5.3 Reasoning Behavior Analysis

We further examine whether Guided Decoding induces meaningful changes in the underlying reasoning process, rather than merely altering answer accuracy. We analyze the generated responses from both quantitative and qualitative perspectives.

**Quantitative error analysis.** We randomly sample 100 questions from GSM8K and generate responses at five representative reasoning simulation levels. We use GPT-4o as an LLM judge to identify and categorize errors in the generated chain-of-thought responses, including logical errors, false prior knowledge, and surface-heuristic mistakes. The resulting annotations are subsequently verified by human annotators, with an overall agreement of 95% between the LLM and human annotations.

As shown in Table 3, the frequency of reasoning errors generally decreases as the simulation level increases. Responses at lower levels contain more logical inconsistencies, heuristic shortcuts, and incorrect assumptions, whereas responses at higher levels exhibit fewer errors and more coherent reasoning. The average number of errors per response also decreases consistently across the evaluated levels. These results indicate that the controlled performance progression is accompanied by systematic changes in reasoning quality, rather than arbitrary perturbations to the generated answers.

**Qualitative analysis.** Table 4 provides a representative example of the reasoning transitions produced by Guided Decoding. The reference response presents a complete and logically coherent solution. At Level 6, the model still reaches the correct answer, but compresses the reasoning into a shorter arithmetic expression, reflecting a simplified, surface-level reasoning strategy. At Level 3, the response becomes structurally inconsistent: the model incorrectly combines two mutually exclusive purchasing options, resulting in a logical error. At Level 1, the reasoning deteriorates further, as the model misapplies basic numerical information and produces an invalid calculation.

Together, the quantitative and qualitative results show that Guided Decoding produces ordered and interpretable transitions in reasoning behavior. Lower simulation levels are associated with increasingly frequent and severe reasoning errors, while higher levels progressively recover coherent and accurate reasoning. This

Table 5: Prompt-source robustness on GSM8K/Mistral. Lower PDS/SDS is better.

| Guidance Source | PDS↓ | SDS↓ |
|---|---|---|
| GPT-4o generated (original) | 9.56 | 4.26 |
| Human (online-collected) | **9.27** | **4.08** |
| Best baseline (betterRP) | 44.07 | 11.72 |

Table 6: Decoding-time controls on RACE/Mistral. Lower PDS/SDS is better.

| Method | PDS↓ | SDS↓ |
|---|---|---|
| Direct good-vs-poor interpolation ($K = 1$) | 238.97 | 44.30 |
| Uniform multi-guidance fusion | 91.99 | 20.20 |
| Temperature-randomness control | 224.79 | 45.50 |
| Guided Decoding | **30.78** | **15.24** |

suggests that Guided Decoding controls not only task performance, but also the quality and structure of the reasoning process.

## 5.4  Robustness and Decoding-Time Controls

**Prompt-source robustness.** To test whether controllability depends specifically on GPT-4o-generated guidance, we replace the GPT-4o-generated guidance prompts with guidance prompts collected from real online human responses that exhibit the same predefined error types: logical errors, false prior knowledge, and surface heuristics. These online-collected responses are not drawn from GSM8K test items or test labels. All other settings are unchanged (Mistral-7B, greedy decoding, full GSM8K test split). As shown in Table 5, Guided Decoding retains comparable controllability, indicating that the effect is not tied to GPT-4o-generated guidance.

**Decoding-time controls.** We also compare against decoding-time controls matched to Guided Decoding on every factor except the fusion design: the same backbone (RACE/Mistral), the same reference/guidance prompts, the same token-level logit access, and the same level schedule $\lambda_l = l/10$. Direct good-vs-poor interpolation uses one reference stream and one guidance stream (the resource-matched contrastive/DExperts-style analogue in our setting, since we have no separately trained expert/anti-expert models); uniform multi-guidance fusion keeps the same $K = 3$ guidance prompts but removes similarity weighting; the temperature-randomness control tests whether simple decoding perturbation alone can produce a smooth proficiency trajectory. Because only the fusion design varies across rows, any gap isolates the contribution of the fusion mechanism rather than token-level access or prompt resources. Table 6 shows that these controls remain substantially worse than full Guided Decoding, supporting the contribution of combining multiple guidance streams with similarity-weighted aggregation.

**Statistical uncertainty.** Because greedy decoding is deterministic for a fixed model, prompt set, and hyperparameters, repeated greedy decoding with the same setup produces the same outputs. We estimate uncertainty by introducing temperature randomness and changing the random seed, then re-running the full 11-level evaluation. Table 7 reports mean PDS/SDS and standard errors across these temperature-randomness seeds.

## 5.5  Generalization Across Model Scales

To examine whether our method generalizes to larger models, we conduct additional experiments on STRATE-GYQA using larger variants from the Mistral and Qwen model families. Specifically, we evaluate Mistral–8B/24B and Qwen–8B/14B/32B under four settings: Zero-shot, Few-shot, BetterRP, and our Guided Decoding method. The results are summarized in Table 8.

Table 7: Standard errors across temperature-randomness seeds. Lower PDS/SDS is better.

| Dataset | Backbone | PDS | PDS SE | SDS | SDS SE |
|---------|----------|-----|--------|-----|--------|
| RACE | Mistral | 30.78 | 4.89 | 15.24 | 1.33 |
| GSM8K | Mistral | 9.56 | 2.03 | 4.26 | 0.98 |
| StrategyQA | Mistral | 19.99 | 4.74 | 8.11 | 2.26 |
| MuTual | Mistral | 36.97 | 6.70 | 14.68 | 4.11 |

Table 8: Results on STRATEGYQA using larger Mistral and Qwen variants. Lower PDS and SDS indicate closer alignment with the target performance trajectory. The best result for each model and metric is shown in bold.

| Model | Zero-shot | | Few-shot | | BetterRP | | Ours | |
|-------|-----------|-----------|----------|----------|----------|----------|------|------|
| | **PDS ↓** | **SDS ↓** | **PDS ↓** | **SDS ↓** | **PDS ↓** | **SDS ↓** | **PDS ↓** | **SDS ↓** |
| Mistral-8B | 67.25 | 18.38 | 74.97 | 23.16 | 38.94 | 13.72 | **19.99** | **8.11** |
| Mistral-24B | 71.03 | 31.65 | 78.68 | 44.82 | **48.35** | 34.17 | 65.75 | **28.65** |
| Qwen-8B | 100.23 | 21.44 | **27.70** | 16.87 | 110.84 | 25.89 | 30.14 | **11.70** |
| Qwen-14B | 60.80 | 16.79 | 98.59 | **14.77** | 118.79 | 19.66 | **25.99** | 16.26 |
| Qwen-32B | 44.19 | 18.87 | 72.82 | 31.04 | 75.36 | 26.96 | **23.11** | **12.36** |

From the results, we observe that scaling the backbone size does not substantially alter the relative performance trends across methods. Across both model families, Guided Decoding remains competitive with prompting-based baselines, and often achieves lower PDS and SDS scores.

In addition, our method maintains stable performance across different model scales. Even as the backbone size increases (e.g., Qwen–14B/32B and Mistral–24B), Guided Decoding continues to produce strong controllability results, reflected in consistently low PDS and SDS values. These observations suggest that the proposed decoding strategy generalizes well across model capacities and remains effective when applied to larger LLM backbones.

### 5.6 Effect of Similarity-Based Weighting

We also ablate the impact of similarity-based weighting when aggregating guidance logits. In our default setting, we compute guidance weights according to the divergence between each guidance probability distribution and the reference probability distribution at each decoding step. These weights assign greater importance to guidance streams that provide more distinct and non-redundant signals relative to the reference trajectory, while the aggregation itself is still performed over raw guidance logits. We compare this to a variant that uses uniform averaging over guidance logits without similarity weighting.

Table 9: Ablation study on the effect of probability-based similarity weighting for guidance aggregation.

| Model | Sim | Performance at Levels 1–9 | | | | | | | | | PDS | SDS |
|-------|-----|------|------|------|------|------|------|------|------|------|-----|-----|
| Mistral | N | 34.47 | 40.53 | 49.51 | 57.31 | 66.50 | 72.84 | 76.32 | 78.69 | 79.59 | 91.99 | 20.20 |
| | Y | 28.13 | 36.42 | 43.45 | 50.00 | 56.82 | 63.85 | 67.90 | 71.03 | 72.91 | **30.78** | **15.24** |
| LLaMA-8b | N | 42.86 | 48.46 | 55.36 | 62.74 | 69.84 | 75.13 | 77.71 | 78.20 | 78.69 | 96.06 | **18.83** |
| | Y | 40.95 | 46.45 | 54.31 | 61.63 | 69.15 | 74.93 | 77.64 | 78.20 | 79.11 | **89.44** | 19.92 |
| LLaMA-70b | N | 14.97 | 22.21 | 34.54 | 56.54 | 72.49 | 85.58 | 89.48 | 91.22 | 91.43 | 117.19 | 50.27 |
| | Y | 14.28 | 21.52 | 33.42 | 54.73 | 71.58 | 84.82 | 88.92 | 90.80 | 91.09 | **112.65** | **49.78** |

Table 10: Ablation study on the number of guidance prompts.

| # Guidance | Performance at Simulated Levels (1 to 9) | | | | | | | | | PDS | SDS |
|---|---|---|---|---|---|---|---|---|---|---|---|
| few-shot | 78.97 | 77.99 | 79.11 | 79.03 | 79.25 | 79.24 | 79.31 | 79.31 | 79.10 | 247.54 | 45.55 |
| 1 | 68.45 | 79.04 | 78.55 | 79.31 | 79.52 | 79.45 | 79.45 | 79.38 | 79.59 | 238.97 | 44.30 |
| 2 | 38.92 | 51.39 | 63.23 | 71.23 | 75.90 | 78.41 | 79.59 | 79.80 | 79.87 | 154.57 | 35.09 |
| 3 | 28.13 | 36.42 | 43.45 | 50.00 | 56.82 | 63.85 | 67.90 | 71.03 | 72.91 | **30.78** | **15.24** |

As shown in Table 9, similarity-based weighting improves PDS across all tested RACE backbones and often improves local smoothness. For example, on Mistral, similarity-based weighting reduces PDS from **91.99** to **30.78** and SDS from **20.20** to **15.24**. On LLaMA-8b and LLaMA-70b, it also improves PDS (**96.06** → **89.44** and **117.19** → **112.65**, respectively), while the SDS change is not uniformly positive.

These results suggest that similarity-aware weighting improves the fusion process by adaptively weighting guidance streams according to their divergence from the reference trajectory. The gains are most significant on smaller models like Mistral, where reasoning is more sensitive to input variation. On stronger models like LLaMA-70b, the gains are more modest, possibly because the model is already capable of filtering incoherent signals internally.

Overall, this ablation validates the role of similarity-aware fusion in improving global alignment and often helping local consistency, while the SDS gains are not uniform across all backbones.

### 5.7 Ablation on Guidance Prompts

To examine how the number of guidance prompts influences controllability, we vary the number of guidance prompts used in Guided Decoding from 1 to 3 and measure performance across reasoning simulation levels on the RACE dataset, as shown in Table 10. The few-shot baseline demonstrates little variation across simulation levels, indicating limited ability to regulate reasoning behavior through prompting alone. In contrast, even with a single guidance prompt, Guided Decoding already produces a measurable performance gradient across levels, resulting in lower PDS and SDS. Adding a second guidance prompt significantly improves controllability. The expanded set of guidance signals introduces more diverse reasoning trajectories, allowing the decoding process to better regulate intermediate behavior and resulting in further reductions in PDS and SDS. With three guidance prompts, the model achieves the best results, exhibiting smoother performance transitions and clearer separation across simulation levels. These results suggest that increasing the diversity of guidance prompts improves the model's ability to modulate reasoning behavior during decoding.

### 5.8 Ablation on Reasoning Prior Scaling

To investigate the effect of moderating the reasoning prior during decoding, we conduct an ablation study on the scaling factor $\beta$ using the RACE dataset with Mistral as the backbone. Table 11 summarizes the results across different $\beta$ values.

We observe that moderate values of $\beta$ (e.g., $\beta = 0.5$) provide the best balance between global alignment and local consistency, yielding the lowest PDS (30.78) and competitive SDS (15.24). When $\beta$ is too small, the influence of the reference reasoning stream is overly suppressed, leading to unstable performance transitions across simulation levels. Conversely, when $\beta$ is too large, the reference reasoning prior becomes dominant, limiting the contribution of guidance signals during logit fusion.

These results suggest that appropriately moderating the reference reasoning prior helps balance the contributions of reference and guidance streams, thereby improving controllability across reasoning simulation levels.

Table 11: Ablation study on the reference scaling factor $\beta$. Moderate values (e.g., $\beta = 0.5$) achieve the best balance between PDS and SDS.

| $\beta$ | 0.3 | 0.4 | 0.5 | 0.6 | 0.7 |
|---|---|---|---|---|---|
| PDS↓ | 74.96 | 34.93 | **30.78** | 65.93 | 107.51 |
| SDS↓ | 17.12 | 13.23 | **15.24** | 16.96 | 20.67 |

# 6 Limitations

Guided Decoding has several limitations. First, it requires token-level (logit) access to the model, so it is not directly applicable to closed-source APIs without such access. Second, it is bounded by the reasoning patterns encoded in the guidance prompts and cannot generate entirely novel reasoning trajectories beyond those signals. Third, our current evidence supports controllable performance modulation and systematic changes in common reasoning-error patterns, but not full fidelity to real learner or user response distributions, which would require validation against real human data. Fourth, compute and KV-cache memory grow with the number of guidance streams; although the streams are batched into a single forward pass, batching does not remove the additional cost.

# 7 Broader Impact

Guided Decoding is designed for educational and research applications where controllable reasoning simulation is beneficial, such as student/user simulation, robustness testing, and evaluation of tutoring systems. However, the ability to controllably generate weaker, plausible-looking but subtly incorrect reasoning on demand carries dual-use risk: such content is exactly the kind that non-experts find hard to catch, and it could be misused to produce misleading tutoring material or assessment items, or be mistaken for real learner data. We therefore recommend the following safeguards: generated outputs should be clearly labeled as synthetic; they should be validated against real learner data before deployment; demographic attributes should not be used as proxies for reasoning ability; induced error patterns should be audited for bias; and high-stakes educational or assessment use should require human review.

# 8 Conclusion

In this work, we formulate the problem of controllable reasoning proficiency simulation in large language models and identify the *Reasoning Prior Bias*, whereby models tend to follow similar high-proficiency reasoning patterns despite being prompted to simulate users with different abilities. To address this limitation, we propose Guided Decoding, a parameter-training-free decoding-time approach that combines reference and guidance logits to continuously modulate the reasoning behavior expressed during generation. Experiments across multiple question-answering benchmarks and model families demonstrate that Guided Decoding generally achieves better global alignment and more consistent transitions across proficiency levels than baselines. Overall, Guided Decoding provides a lightweight approach for controllably modulating reasoning behavior without model fine-tuning. We emphasize that our evidence supports controllable performance modulation and systematic changes in common reasoning-error patterns; establishing full fidelity to real student or user response distributions requires validation against real human data and remains an important direction for future work.

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

# A  Dataset

To cover a comprehensive range of subject for student simulation. We experiment on four datasets. The **StrategyQA** dataset (Geva et al., 2021) is a question-answering benchmark focusing on opendomain questions, requiring implicit reasoning to infer the necessary steps from the question itself through a strategic approach. It is designed to evaluate the ability to perform implicit reasoning, necessary for answering questions that do not have direct or explicit answers within the text.

The **GSM8K** dataset (Cobbe et al., 2021) is structured to facilitate question answering on fundamental mathematical problems that require multi-step reasoning for resolution. The solutions to these problems primarily involve performing a sequence of elementary calculations using basic arithmetic operations.

The **RACE** dataset (Lai et al., 2017) is a multiple-choice question answering benchmark derived from English reading comprehension exams. It tests the ability to comprehend complex passages, extract relevant information, and answer questions that require inferential and contextual understanding, making it ideal for simulating reading-based academic tasks.

The **MuTual** dataset (Cui et al., 2020) is a multiple-choice benchmark for multi-turn dialogue reasoning, constructed from English listening comprehension exams. Each instance presents a dialogue context with subtle distractors and requires selecting the most appropriate next utterance from several candidates. Unlike single-turn QA datasets, MuTual emphasizes pragmatic understanding, dialogue coherence, and reasoning over conversational flow, making it a challenging testbed for simulating interactive reasoning and communication proficiency.

## A.1  Reasoning Diversity Evaluation

To directly assess the variety of reasoning errors generated by our method, we conducted a targeted evaluation using LLM-as-a-Judge (Zheng et al., 2023). We randomly sampled 100 incorrect responses produced on the GSM8K dataset. Each response was then analyzed using GPT-4o to classify the dominant error type(s) according to our predefined taxonomy. The distribution of error types was as follows: 44% of responses contained logical errors, 22% reflected false prior knowledge, and 34% were attributed to surface heuristics. Additionally, we asked GPT-4o to assess whether multiple error types occurred within a single response. The analysis revealed that 19% of the responses exhibited two or more error types, indicating that our method can simulate compound or mixed reasoning failures.

We also provide a qualitative example: *Question: Raymond and Samantha are cousins. Raymond was born 6 years before Samantha. Raymond had a son at the age of 23. If Samantha is now 31, how many years ago was Raymond's son born?*

*Model Answer: Raymond was born 6 years before Samantha. So Raymond is 6 years older than Samantha. Raymond had a son at the age of 23. So Raymond's son is 23 years old. Samantha is now 31. So Raymond's son is 31 - 23 = 8 years old.*

This response exhibits both a logic error, incorrectly aligning the timeline of Raymond's age with Samantha's, and a surface heuristic, namely the use of direct subtraction between salient numbers without temporal justification.

# B  Ablation Study: Number of Guidance Prompts

In the main paper, we reported that using three guidance prompts produced the strongest performance *among the evaluated settings*. Our intention was not to claim that three prompts is globally optimal, but rather that it was the best choice within the tested configuration. To further clarify this point, we extend the ablation to include four and five guidance prompts. The complete results are presented in Table 12.

The results reaffirm that using three guidance prompts is a strong and efficient default choice. While increasing the number of prompts to four or five yields comparable performance, the improvements over three prompts are marginal. Additionally, for resource-constrained scenarios, even two prompts provide substantial gains

| # Prompts | PDS $\downarrow$ | SDS $\downarrow$ |
|---|---|---|
| 1 | 238.97 | 44.30 |
| 2 | 154.57 | 35.09 |
| 3 | **30.78** | **15.24** |
| 4 | 37.81 | 15.45 |
| 5 | 37.96 | 15.24 |

Table 12: Ablation on the number of guidance prompts. Lower PDS and SDS indicate better proficiency alignment.

relative to one prompt or zero guidance baseline methods. Overall, three prompts offer an effective balance between controllability, stability, and computational cost.

## C    Inference Overhead Justification

While batching improves GPU utilization, Guided Decoding still evaluates one reference stream and $K$ guidance streams, so compute and KV-cache memory scale with the number and length of these streams. We therefore report wall-clock and memory measurements rather than describing the overhead as negligible. In our Mistral-7B bf16 measurement over 886 RACE samples, standard greedy decoding ($K = 0$, one stream) takes 718s in total (0.81s/sample, 74.04 samples/min, 16.3GB peak memory). Unbatched $K = 3$ decoding uses four logical streams and takes 2609s (2.94s/sample, 20.38 samples/min, 16.4GB peak memory). Batched $K = 3$ decoding keeps the same four logical streams but executes them as one batch, taking 0.84s/sample with 18.2GB peak memory. Thus batching substantially reduces wall-clock latency, while memory and total computation still reflect the additional guidance streams.

## D    Prompts

```
Prompt Template for GPT

### **A (Expert / High Mastery)**

* Solves **multi-step** and **non-routine** math problems accurately.
* Uses **correct reasoning**,checks constraints/edge cases,and can **prove/justify**
steps.
* Chooses efficient strategies (algebraic manipulation,casework,invariants,etc.).
* Rarely makes calculation mistakes; can self-correct quickly.

### **B (Strong / Proficient)**

* Solves most **standard** math problems correctly,especially with familiar patterns.
* Reasoning is mostly correct but may miss **one tricky step** or forget a condition.
* Can handle multi-step problems,but is less confident on **novel** or proof-style
questions.
* Occasional arithmetic/algebra slips; usually fixable with checking.

### **C (Intermediate / Developing)**

* Can solve **basic to medium** problems but struggles on longer chains of reasoning.
* Often applies formulas mechanically without fully understanding why they work.
```

```
* Makes common mistakes (sign errors,wrong simplification,missing cases).
* Needs hints like "try substitution," "draw a diagram," or "break into cases."

### **D (Beginner / Limited)**

* Struggles with **foundational skills** (operations,algebra manipulation,equations).
* Frequently cannot set up the correct approach or gets stuck early.
* Reasoning is incomplete; guesses steps or uses incorrect formulas.
* Requires step-by-step guidance and practice with core concepts.

You are an role student. Please answer the question: question
```

```
Prompt Template for RACE

You will be shown a multiple choice question from an English reading comprehension
exam. You must assign a difficulty level to the given multiple choice question,and
select the answer choice that a student would pick. Here are some examples.

## Example 1 ##

{example_question}

You must assign a difficulty level to the given multiple choice question,and select
the answer choice that a **good** level student would pick.

Please first think about the reasoning process that a **good** level student would
follow to select the answer,including the misconceptions that might cause them to
make mistakes. And then give me the answer. Format: Reasoning process. <Answer>your
choice</Answer>

**Your Answer:**

{example_response_good}

## Example 2 ##

{example_question}

You must assign a difficulty level to the given multiple choice question,and select
the answer choice that a **poor** level student would pick.

Please first think about the reasoning process that a **poor** level student would
follow to select the answer,including the misconceptions that might cause them to
make mistakes. And then give me the answer. Format: Reasoning process. <Answer>your
choice</Answer>

**Your Answer:**

{example_response_bad}
```

```
## Example End ##

Reading passage: "{context}"

Question: "{question}"

Options: "{option}"

You must assign a difficulty level to the given multiple choice question,and select
the answer choice that a **{student_level}** level student would pick.

Please first think about the reasoning process that a **{student_level}** level
student would follow to select the answer,including the misconceptions that might
cause them to make mistakes. And then give me the answer. Format: Reasoning process.
<Answer>your choice</Answer>

**Your Answer:**
```

```
RACE Example Question

Reading passage: "People who are about the same age as you are your peers. Peers
include your friends and your classmates. They have strong influence on your actions.
Your peers influence how you think,how you act,and even how you dress. Peer pressure
is the influence that people of the same age have on one another.
Sometimes your peers influence you in a helpful or positive way. A friend may notice
that you have problems in math. And he might invite you to join a study group or show
you how to solve a difficult problem during lunch. Such actions are helpful to you.
Sometimes your peers influence you in a  or unhealthy way. A friend might offer
you cigarettes .Cigarettes are harmful to your health. Your friend knows that. Your
friend also knows that underage smoking is against the law. Yet he or she still makes
the offer. This bad influence is negative peer pressure.
Your peers may not always realize they are influencing you in a negative way. For
example. a friend might invite you to the movies. You would love to go,but you have a
lot of housework to do. In situations like this you should make a wise decision.
You can learn to deal with negative peer pressure. Keep away from people who try to
influence your behavior in harmful ways. Though it is not always easy to say no,itś
really necessary to learn to do that. Follow the following steps. First. look at the
person and say no in a firm but polite voice. Make sure your face shows that you are
serious. Let the person know you will not back down. Then,give reasons for saying no.
Explain why you wont́ do what the person wants. Remember to say goodbye and walk away
if he or she continues."
Question: "Which sentence shows the writerś opinion?"
Options: "A: Peer pressure is the influence that people of the same age have on one
another.
B: We should learn to deal with negative peer pressure.
C: Itś not always important to say no to your peers.
D: Peers include your friends and your classmates.
"
```

RACE Reference Example Answer

<Answer>B</Answer>
Reasoning process:
The student would recognize that the question is asking for the writer's opinion,which is different from a factual statement. They would analyze each option and identify which one contains a subjective perspective or a recommendation. Option A is a factual definition of peer pressure,and options C and D also present factual or descriptive statements. However,option B includes the word "should," which indicates advice or guidance. This suggests that the writer is sharing their opinion on how to handle negative peer pressure. Therefore,a good student would select option B as the correct answer.answer.

RACE Guidance Example Answers

<Answer>A</Answer>
Reasoning process:
The student might struggle to distinguish between facts and opinions. They might think that since the sentence provides a clear and straightforward definition of peer pressure,it must be the writer's opinion. They could focus on the literal meaning of the sentence rather than evaluating whether it expresses a subjective viewpoint or advice. The student may overlook the importance of words like "should" in option B,which actually signals the writer's opinion,and instead choose a more neutral or general statement that seems important in the passage.

<Answer>C</Answer>
Reasoning process:
The student might misunderstand the passage due to unfamiliar vocabulary like "peer pressure," "influence," or "negative." They may pick option C because the phrase "not always important" sounds like advice or something the writer could be saying. Without fully understanding the context or meaning of the sentence,they might guess based on the tone of the words,not realizing that option C actually contradicts the writer's main message. Limited vocabulary comprehension leads the student to focus on surface-level language rather than meaning.

<Answer>D</Answer>
Reasoning process:
The student might simply choose the first sentence they remember from the passage without thinking about whether it shows an opinion. They may confuse something that *sounds like it came from the writer* with something that *is the writer's opinion*. Since the sentence in option D talks about who peers are,the student might think,"Well,the writer said that,so it must be their opinion." This reflects a misunderstanding of the difference between stating a fact and expressing a viewpoint. The student is likely relying on memory rather than critical reading.

**StrategyQA Example Question**

Question: "Is a Boeing 737 cost covered by Wonder Woman (2017 film) box office receipts?" Options: "A: True B: False"

---

**StrategyQA Reference Example Answer**

<Answer>A</Answer>
Reasoning process:
The average cost of a US Boeing 737 plane is 1.6 million dollars. Wonder Woman (2017 film) grossed over 800 million dollars at the box office. So,the Answer is A.

---

**StrategyQA Guidance Example Answer**

<Answer>B</Answer>
Reasoning process:
Making a movie costs a lot of money. Since Wonder Woman probably spent millions to make,there's no way the movie made enough profit to cover a plane like a Boeing 737. So the answer is B.

<Answer>B</Answer>
Reasoning process:
A Boeing 737 costs over a billion dollars,and movies don't make that much,so Wonder Woman couldn't cover the cost. Therefore,the answer is B.

<Answer>B</Answer>
Reasoning process:
Wonder Woman is a movie character,and she doesn't buy airplanes. So she would not cover the cost of a Boeing 737. Therefore,the answer is B.

---

**Mutual Guidance Example Question**

f : hi ,jack . could you help me carry these books ? m : i 'd love to ,mary ,but i need to meet professor johnson in his office immediately . Options: A: "f : you are going to your own office immediately .",B: "f : you will help me carry the books first . thank you .",C: "f : you are going to help me carry these books before you go to professor johnson 's office immediately . you are so nice .",D: "f : you are going to professor johnson immediately ."""

---

**Mutual Reference Example Answer**

<Answer>D</Answer>
Reasoning process:

27

```
The female speaker (f) should acknowledge his reason for declining. Option D
is a simple,logical paraphrase of his statement,showing understanding without
contradicting him.
So,the Answer is D.
```

```
Mutual Guidance Example Answer

<Answer>C</Answer>
Reasoning process:
The female speaker (f) doesn't care if Jack has somewhere to be,he can help me first.
It's rude of him to just walk away when I'm clearly struggling.
So,the Answer is C.
```

