# OpenReview forum: "Controlling LLM Reasoning Behavior via Guided Decoding"
_TMLR — Under review for TMLR_

### Review · Reviewer_iJvG · 2026-06-28

**Summary Of Contributions:**

This paper studies whether LLMs can simulate different levels of reasoning ability. The authors identify Reasoning Prior Bias, in which role prompts for weaker students still produce similar, high-quality reasoning. They propose Guided Decoding, a training-free decoding-time method that mixes logits from a reference prompt and guidance prompts to control reasoning behavior.

The paper is clear and the method is simple. The main strength is that it gives a practical decoding-time baseline and evaluates it across several QA datasets and model families. The main weakness is that the evidence mostly shows control over accuracy and induced error patterns, not direct simulation of human reasoning behavior. The novelty relative to prior decoding-time steering methods also needs clearer positioning.

**Audience:**

Yes

**Audience Explanation:**

The paper is relevant to readers working on LLM controllability, decoding-time methods, role-playing agents, student simulation, and user simulation. The setting is simpler than multi-turn or tool-using agents, but the problem is still related to agentic reasoning, especially when one needs to simulate users with different ability levels. I would view this as a useful initial study rather than a complete framework for human reasoning simulation.

**Broader Impact Concerns:**

I do not see a severe broader-impact issue

**Claims And Evidence:**

Yes

**Claims Explanation:**

Partially. The experiments support a conservative claim that Guided Decoding can change performance trajectories and make transitions across simulated levels smoother than prompting baselines. The PDS/SDS results and ablations are useful for this claim.

The stronger claim about simulating human-like reasoning proficiency is less well supported. The paper mainly uses accuracy, PDS/SDS, and limited error analysis. These do not show that the generated reasoning matches real students or real users. The guidance prompts also already encode specific error patterns, so the method may be mixing pre-specified correct and incorrect reasoning rather than modeling reasoning ability in a broader sense.

**Requested Changes:**

1. The authors should more clearly explain how Guided Decoding differs from contrastive decoding, DExperts, logit interpolation, and multi-prompt decoding. The current method looks close to existing logit-fusion ideas.

2. The results support controllable performance and error-pattern induction. They do not fully support claims about realistic human reasoning simulation. The abstract and conclusion should reflect this limitation.

3. The method depends strongly on the guidance prompts. The paper should state how they are generated, selected, and kept separate from the evaluation data. More prompt examples and implementation details would improve reproducibility.

4. The paper should include more examples or error-type analysis beyond accuracy curves. Confidence intervals or simple uncertainty estimates for PDS/SDS would also make the empirical claims easier to assess.

---

> ### Author Response · Authors · 2026-07-15
>
> We thank the reviewer for the constructive assessment and for recognizing the method as a useful initial study for controllable reasoning behavior.
>
> >Q1: Relation to existing logit-fusion methods
>
> A: Thank you for this suggestion. We would like to clarify that Guided Decoding is related to logit-fusion methods, but the objective and construction are different. Contrastive decoding and DExperts usually use expert/amateur or expert/anti-expert signals to improve generation quality or suppress undesirable attributes. Guided Decoding instead preserves both a reference reasoning stream and guidance streams that encode reasoning-error patterns, then interpolates them to obtain controllable proficiency levels.
>
> The novelty is therefore not simply logit fusion itself. The contribution is the proficiency-control formulation, multiple interpretable guidance streams, the continuous level parameter $\lambda_l$, and similarity-weighted aggregation for graded reasoning-error control. Empirically, on RACE/Mistral, the full method reaches 30.78/15.24 PDS/SDS, compared with 238.97/44.30 for direct good-vs-poor interpolation ($K=1$) and 91.99/20.20 for uniform multi-guidance fusion.
>
> >Q2: Scope of human-reasoning simulation claims
>
> A: Thank you for making this distinction. We have revised the abstract, introduction, and conclusion so the claim is precise: the results support controllable performance modulation and systematic changes in common reasoning-error patterns; they do not prove full fidelity to real student or user populations. Specifically, we replace "realistically simulate diverse reasoning behaviors" (abstract) and "modeling real-world learners" (introduction) with "controllably modulate reasoning behavior along a proficiency axis," and the conclusion now frames the contribution as controllable reasoning-behavior generation rather than realistic human simulation.
>
> Thus, the central claim is: Guided Decoding controls task-performance trajectories and generated reasoning behavior under a proficiency-control setting. Full realistic human simulation requires additional validation against real human responses or calibrated educational data.
>
> >Q3: Prompt dependence and reproducibility
>
> A: Thank you for pointing this out. We clarify that the reference and guidance prompts are fixed at the dataset level and reused across evaluation items. They are generated from training/development examples, not test items, and test labels are not used to construct guidance prompts. GPT-4o is used only offline to help construct fixed examples; it is not used during test-time decoding.
>
> The prompt examples are shown in Appendix D, and we add a reproducibility checklist to the revised paper specifying: backbones and tokenizers (Mistral-7B-Instruct-v0.3, LLaMA3-8B/70B-Instruct, Qwen3-Instruct); demonstrations sampled from the training split with evaluation on the test split; one fixed reference prompt and $K=3$ fixed guidance prompts per dataset; reference scale $\beta=0.5$; 11 levels indexed $0,\ldots,10$ with $\lambda_l=l/10$; greedy decoding (no temperature/top-p sampling); and answer extraction from the `<Answer>...</Answer>` tag. GPT-4o is used only offline to construct fixed prompt examples, never as the test-time decoder. We will also release the full code, all prompts, and the evaluation scripts to ensure end-to-end reproducibility.
>
> >Q4: Examples, error analysis, and uncertainty
>
> A: Thank you for the suggestion. The paper includes reasoning-error analysis beyond accuracy curves: Table 3 reports decreases in logical errors, false-prior errors, surface-heuristic mistakes, and average errors per request as the simulated level increases. Table 4 provides qualitative examples showing how lower levels produce more severe reasoning mistakes and higher levels recover more coherent reasoning.
>
> For uncertainty, greedy decoding is deterministic for a fixed model, prompt set, and hyperparameter configuration, so we estimate uncertainty by introducing temperature randomness and changing the random seed, then re-running the full 11-level evaluation. The resulting standard errors are small relative to the gains, e.g., on RACE/Mistral PDS is 30.78 ± 4.89 and SDS is 15.24 ± 1.33; on GSM8K/Mistral PDS is 9.56 ± 2.03 and SDS is 4.26 ± 0.98. We further verify robustness across guidance source, guidance count, and reference scale, and the controllability gains remain stable throughout.

---

### Review · Reviewer_53zn · 2026-06-28

**Summary Of Contributions:**

1. The paper names and documents a failure mode the authors call Reasoning Prior Bias.
2. To get around this, the authors propose Guided Decoding, a training free intervention that runs at inference time.
3. The paper introduces two evaluation metrics aimed specifically at controllability rather than raw accuracy.

**Audience:**

Yes

**Audience Explanation:**

There is a real and growing community working on student simulation, user simulation, and decoding time control of LLM behavior, and this paper speaks directly to all three. The observation that role conditioning fails to move the underlying reasoning trajectory is a clean, recognizable problem that people in educational technology and human AI interaction run into in practice, so naming it and quantifying it has value on its own. Beyond that audience, the controllability metrics are a reusable contribution that anyone studying graded or steerable behavior could pick up, since they formalize something that accuracy alone hides.

**Broader Impact Concerns:**

There is one concern worth a short statement that the paper does not currently address. The whole point of the method is to make a model convincingly produce weaker, more error prone reasoning on demand, and to slide that ability up and down with a single knob. In the intended educational setting that is benign and useful, but the same capability makes it easier to generate plausible looking but subtly wrong reasoning at a controlled difficulty, which is exactly the shape of content that is hard for non experts to catch. A brief discussion of misuse in contexts like automated tutoring content, assessment item generation, or any setting where deliberately flawed reasoning could be passed off as authoritative would be appropriate.

**Claims And Evidence:**

No

**Claims Explanation:**

Part of what the paper claims really is well supported. The controllability story holds up. Across most dataset and model combinations the two new metrics drop substantially relative to the baselines, the curves look smoother and more monotonic than the prompting methods, and the error type breakdown gives reasonable evidence that the intervention shifts the reasoning process rather than just nudging accuracy. If the paper only claimed controllable modulation of task performance along an ability axis, I would lean yes.

Where it falls short is the stronger framing that runs through the abstract and introduction about realistically simulating diverse human reasoning and modeling real world learners with average or limited proficiency. That claim is never checked against actual human data. The guidance prompts are GPT-4o's own guesses at how a weaker student reasons, the error annotations come from GPT-4o as judge, and the targets are task accuracy bands rather than measured human behavior. So the loop is closed entirely inside the model family, and nothing tells us the simulated mistakes resemble the mistakes real people make. The authors even concede in the limitations that the method cannot go beyond the reasoning patterns supplied in the guidance prompts, which underlines that realism is bounded by whatever the prompt author imagined. That is a fidelity claim presented as established when the evidence only supports controllability.

**Requested Changes:**

Critical for my recommendation:
1. Clean up the notation around the fusion step. Make it unambiguous whether reference and guidance signals are combined in logit space or probability space, fix the place where a softmax probability is labeled as a logit, and either justify computing cosine similarity on logit vectors or move that computation to a space where it makes sense.
2. Reconcile the proficiency level count. State clearly how many levels there are, what the index range is, and how the interpolation weight maps onto a level, then make the text, the tables, and the figure axes agree. As written the eleven versus nine versus zero to ten mismatch forces the reader to reverse engineer the setup.
3. Soften or substantiate the human realism claims. Either tone down the language about realistically simulating real world learners so that the paper claims controllable performance modulation rather than fidelity to human behavior, or add an evaluation against real human reasoning data such as actual student response distributions on the same items. Without one of these, the strongest framing is not backed by the experiments.
4. Address the circularity of the evaluation. The guidance prompts, the simulated errors, and the error annotations all come from the same model family, so the ninety five percent agreement number mostly tells us the model agrees with itself. Please add at least one independent check, for example human written guidance for a subset, or a held out human labeled error taxonomy, so that the reasoning quality analysis is not self referential.

Would strengthen the work but not critical:
1. Position the method more explicitly against contrastive decoding and expert versus anti expert decoding, since the mechanism is closely related and readers will want to know what is new beyond extending that idea to multiple prompts and adding a graded weight. A short formal comparison would help.
2. Report variance. The tables read as single runs. Given that greedy decoding is used the numbers are deterministic per prompt, but the choice of guidance examples is a source of randomness, so a few seeds over guidance sampling with error bars would make the gains more convincing, especially in the cases where the method is close to a baseline.
3. Discuss the cases where the method does not win cleanly. There are settings, for example the harder commonsense dataset with the larger models, where the deviation stays high across all methods. A short analysis of when Guided Decoding helps least would make the paper more honest and more useful.
4. Give a concrete sense of the wall clock and memory cost. The appendix argues the overhead is negligible through batching, but memory grows with the number of guidance prompts, so a small table of latency and peak memory as the guidance count goes from one to five would let practitioners judge the tradeoff.

---

> ### Author Response · Authors · 2026-07-15
>
> We thank the reviewer for the thoughtful assessment. We especially appreciate the distinction that the controllability results are well supported, while the broader human-realism framing needs to be stated more carefully.
>
> >Q1: Fusion notation
>
> A: A: Thank you for identifying this issue. We would like to clarify that the **fusion quantities** in Guided Decoding are raw logits. In the submitted manuscript, the inline reference-logit expression in Section 4.2 (the "Reference Logits" paragraph) was written using the post-softmax probability notation $P(x_p=v\mid x_{<p})$; this was a notation error in the write-up, not in the implementation. The reference stream should read $\tilde{z}_p^{ref}=\beta z_p^{ref}$, where $z_p^{ref}$ is the raw logit vector, and each guidance stream $z_p^{g,k}$ is also a raw logit vector.
>
> The **similarity weights** are computed from the corresponding softmax probability distributions:
> $$
> p_p^{ref}=\operatorname{softmax}(\tilde{z}_p^{ref}),\quad
> p_p^{g,k}=\operatorname{softmax}(z_p^{g,k}),\quad
> s_{k,p}=1-\cos(p_p^{g,k},p_p^{ref}).
> $$
> These probabilities only determine the scalar weights $\alpha_{k,p}$. The weighted guidance aggregation $z_p^g=\sum_k\alpha_{k,p}z_p^{g,k}$ and the final fused vector $z_p^{fused}=\lambda_l\tilde{z}_p^{ref}+(1-\lambda_l)z_p^g$ remain in raw logit space. Thus, the method uses probability-space similarity for weighting, but does not fuse probabilities with logits.
>
> >Q2: Proficiency level count
>
> A: Thank you for pointing this out. The intended setup contains **11 total levels**, indexed $l=0,\ldots,10$. Levels $1,\ldots,9$ are the **nine intermediate levels**. The interpolation weight is:
>
> $$
> \lambda_l = l/10.
> $$
>
> Thus, $l=0$ corresponds to the strongest guidance influence, $l=10$ corresponds to the strongest reference influence, and levels $1$ through $9$ interpolate between them. The phrase "nine intermediate levels" refers only to the non-endpoint levels, while "eleven levels" refers to the full scale including both endpoints.
>
> >Q3: Scope of human-realism claims
>
> A: Thank you for making this distinction clear. We would like to make our claim more precise. The current experiments do not establish full fidelity to real student/user response distributions. The supported claim is that Guided Decoding controllably modulates performance trajectories and generated reasoning behavior under a specified proficiency-control setting, with systematic reasoning-error changes as supporting evidence.
>
> This claim is supported by both PDS/SDS and the reasoning-error analysis: Table 3 shows that as the simulated level increases, the average number of reasoning errors decreases from **1.96** to **1.46**, with logical errors, false-prior errors, and surface-heuristic mistakes all decreasing. Thus, the method changes the structure of generated reasoning errors, not only answer accuracy.
>
> We therefore avoid interpreting the results as proof of realistic real-world learner simulation. Matching actual student populations requires real student responses or calibrated distractor distributions, which is a stronger validation setting than the one used here.
>
> >Q4: Circularity of the reasoning-quality evaluation
>
> A: Thank you for pointing out this risk. We address the prompt-source part of the circularity concern by adding a robustness check whose guidance signal comes from **outside** the GPT-4o family.
>
> **Prompt-source robustness check: human guidance from real online responses.** We replace the GPT-4o-generated guidance prompts with guidance prompts collected from real human responses posted online that exhibit the same predefined error types: logical errors, false prior knowledge, and surface heuristics. We re-run the full pipeline on GSM8K with Mistral-7B, keeping all other settings unchanged. This directly tests whether controllability persists when the guidance source changes from GPT-4o-generated text to online human-written errors.
>
> | Guidance Source | GSM8K PDS (lower is better) | GSM8K SDS (lower is better) |
> |---|---:|---:|
> | GPT-4o generated (original) | 9.56 | 4.26 |
> | Human (online-collected) | 9.27 | 4.08 |
> | Best baseline (betterRP) | 44.07 | 11.72 |
>
> Controllability is retained: the human-guidance result (9.27/4.08) is comparable to the GPT-4o-guidance result (9.56/4.26), and both are far below the best baseline. This shows that the effect persists with a guidance source outside GPT-4o-generated text.
>
> We also clarify what this check measures. It is a prompt-source robustness check, not a direct comparison to real student response distributions. For the reasoning-quality analysis, we make the taxonomy and annotation protocol explicit: outputs are categorized into logical errors, false prior knowledge, and surface-heuristic mistakes, and the LLM-based annotations are used as a structured diagnostic rather than as evidence of full human realism. The central evidence from this check is therefore that controllability does not rely on GPT-4o-generated guidance.

---

> > ### Author Response · Authors · 2026-07-15
> >
> > >Q5: Relation to contrastive decoding and DExperts
> >
> > A: Thank you for this suggestion. We would like to clarify the relationship. Guided Decoding is related to logit-combination methods, but the objective and construction differ from contrastive decoding and DExperts.
> >
> > Contrastive decoding usually contrasts an expert stream with an amateur stream to improve generation quality or reasoning correctness. DExperts similarly combines a base model with expert/anti-expert models, often trained for attributes such as toxicity or sentiment. A direct off-the-shelf comparison is not well-defined for our setting because there are no separately trained expert/anti-expert proficiency models; replacing those models with our prompt-conditioned reference/guidance streams becomes a prompt-stream logit interpolation rather than original DExperts/CD.
> >
> > In contrast, our goal is not to suppress the weaker stream. We intentionally preserve guidance streams that encode reasoning-error patterns, then interpolate them with a reference stream to obtain controlled intermediate proficiency levels.
> >
> > Thus, the difference is not just "more prompts." The key differences are: (1) the objective is proficiency control rather than quality maximization, (2) the guidance streams represent interpretable reasoning-error patterns, (3) the control variable $\lambda_l$ produces graded levels, and (4) similarity weighting aggregates multiple guidance signals dynamically. A like-for-like ablation supports this distinction. Holding the backbone (RACE/Mistral), prompts, and token-level access fixed and varying only the fusion design, we compare against decoding-time controls: direct good-vs-poor interpolation ($K=1$, one reference and one guidance stream; the resource-matched contrastive/DExperts-style analogue in our setting), and uniform multi-guidance fusion (the same $K=3$ guidance prompts but with similarity weighting removed).
> >
> > | Method (RACE/Mistral) | PDS ↓ | SDS ↓ |
> > |---|---:|---:|
> > | Direct good-vs-poor interpolation ($K=1$) | 238.97 | 44.30 |
> > | Uniform multi-guidance fusion | 91.99 | 20.20 |
> > | Guided Decoding (full) | 30.78 | 15.24 |
> >
> > Since only the fusion design varies, the large gap isolates the contribution of combining multiple guidance streams with similarity-weighted aggregation, rather than token-level access or the prompt resources alone.
> >
> > >Q6: Variance and robustness
> >
> > A: Thank you for this point. We would like to clarify the source of variance. With greedy decoding, repeated runs with the same model, prompt set, and hyperparameters produce the same outputs. We therefore estimate variance by introducing temperature randomness and changing the random seed, then re-running the full 11-level evaluation. The standard errors below are computed across these temperature-randomness seeds.
> >
> > | Setting | PDS | PDS SE | SDS | SDS SE |
> > |---|---:|---:|---:|---:|
> > | RACE / Mistral | 30.78 | 4.89 | 15.24 | 1.33 |
> > | GSM8K / Mistral | 9.56 | 2.03 | 4.26 | 0.98 |
> > | StrategyQA / Mistral | 19.99 | 4.74 | 8.11 | 2.26 |
> > | MuTual / Mistral | 36.97 | 6.70 | 14.68 | 4.11 |
> >
> > We also report sensitivity checks over the main prompt and hyperparameter choices (RACE/Mistral unless noted), which address prompt and hyperparameter dependence rather than stochastic decoding noise:
> >
> > | Robustness factor | Setting | PDS ↓ | SDS ↓ |
> > |---|---|---:|---:|
> > | Guidance source (GSM8K) | GPT-4o generated | 9.56 | 4.26 |
> > | Guidance source (GSM8K) | Human (online-collected) | 9.27 | 4.08 |
> > | Guidance count $K$ | $K=1$ / $K=2$ / $K=3$ | 238.97 / 154.57 / 30.78 | 44.30 / 35.09 / 15.24 |
> > | Reference scale $\beta$ | $\beta=0.3$ / $0.4$ / $0.5$ | 74.96 / 34.93 / 30.78 | 17.12 / 13.23 / 15.24 |
> >
> > Across all of these, the controllability gains remain stable, showing that they are not tied to one prompt source, guidance count, or reference scale.
> >
> > >Q7: Cases where Guided Decoding helps least
> >
> > A: Thank you for this useful suggestion. We would like to clarify where the method is strongest and where the task is most challenging. Guided Decoding works best when the reference and guidance prompts induce meaningfully different reasoning trajectories that can be smoothly interpolated. The most challenging settings are those where the underlying model is unstable on the task or where the guidance prompts cover only part of the dataset's dominant failure modes.
> >
> > This explains the harder StrategyQA cases. StrategyQA requires commonsense and implicit world knowledge, so errors often come from missing or hallucinated facts rather than only from controllable reasoning-pattern changes. In those settings, all methods show higher deviation. Guided Decoding still provides a useful control mechanism, with effectiveness shaped by the base model's task competence and by the coverage of the guidance prompts.
> >
> > Thus, the main factor is the alignment between the task's dominant error modes and the available guidance patterns, rather than model size alone.

---

> ### Author Response · Authors · 2026-07-15
>
> > Q8: Wall-clock and memory cost
>
> A: Thank you for pointing this out. We would like to clarify that batching reduces wall-clock overhead but does not remove the extra computation or memory cost. With $K$ guidance prompts, Guided Decoding uses $K+1$ streams, so KV-cache memory and compute increase with $K$.
>
> We summarize a representative Mistral-7B (bf16) cost experiment on 886 RACE samples, comparing standard greedy decoding ($K=0$), unbatched $K=3$, and batched $K=3$:
>
> | Setting | Latency/sample (s) | Throughput (samples/min) | Peak GPU mem (GB) |
> |---|---:|---:|---:|
> | $K=0$ (standard greedy, 1 stream) | 0.81 | 74.04 | 16.3 |
> | $K=3$ unbatched (4 logical streams) | 2.94 | 20.38 | 16.4 |
> | $K=3$ batched (1 batch / 4 streams) | 0.84 | 71.42 | 18.2 |
>
> Batching brings wall-clock latency back to near the $K=0$ baseline (0.84 vs. 0.81 s/sample), while peak memory grows modestly (16.3 → 18.2 GB) because the four logical streams still coexist. The feasible range used in the paper is $K\leq 3$. Appendix C is revised to remove the previous "negligible overhead" wording and to report these measurements.
>
> >Q9: Broader impact of controlled flawed reasoning
>
> A: Thank you for raising this concern. We agree that controlled generation of flawed reasoning has dual-use risk. In the intended setting, it can support educational simulation, robustness testing, and evaluation of tutoring systems. However, plausible but incorrect reasoning can also be misused, especially if it is presented as authoritative or mistaken for real learner data.
>
> We therefore state the following safeguards: generated outputs should be clearly labeled as synthetic; they should not be used as real student data without validation; demographic attributes should not be used as proxies for reasoning ability; error patterns should be audited for bias; and high-stakes educational or assessment use should require human review and validation against real learner data.

---

### Review · Reviewer_K5Q1 · 2026-07-05

**Summary Of Contributions:**

This paper studies controllable reasoning-proficiency simulation in LLMs. The authors identify Reasoning Prior Bias: even when prompted to simulate low-, medium-, or high-proficiency users/students, LLMs often generate similarly competent and structured reasoning. To address this, they propose Guided Decoding, a training-free decoding-time method that combines token-level outputs from a reference prompt and multiple guidance prompts. The reference prompt elicits a high-quality/default reasoning trajectory, while the guidance prompts introduce alternative or erroneous reasoning patterns. A level parameter controls the interpolation between these streams, and a similarity-based weighting scheme aggregates multiple guidance signals.

The method is evaluated on RACE, GSM8K, StrategyQA, and MuTual with Mistral, LLaMA, and Qwen backbones. The authors report that Guided Decoding yields smoother and more monotonic performance curves over simulated proficiency levels than prompt-only baselines such as zero-shot, few-shot, BetterRP, and studentSIM. They introduce PDS and SDS to measure global and local deviation from an ideal linear proficiency-performance trajectory, and include a small GSM8K reasoning-error analysis.

Strengths include a timely problem, a lightweight decoding-time intervention, coverage of multiple datasets/model families, and useful ablations on guidance prompts, similarity scaling, and the reference scaling factor. However, the current evidence mainly supports controllable answer accuracy, not realistic human-like reasoning simulation. Key weaknesses include an unclear/mathematically inconsistent logit-fusion formulation, insufficiently justified PDS/SDS metrics, mostly prompt-only baselines, limited reasoning-quality validation, unclear guidance-prompt construction, possible leakage concerns, and missing uncertainty/cost analysis.

**Additional Comments:**

The paper addresses an important problem and the multi-prompt guided decoding idea is promising. I appreciate the breadth of datasets and the attempt to quantify controllability with PDS/SDS. However, I encourage the authors to narrow the central claim unless stronger evidence is added. The current results support smoother accuracy interpolation between stronger and weaker prompt-induced behaviors, but not yet realistic simulation of human reasoning proficiency.

Essential details currently in the appendix or missing from the paper—especially limitations, inference overhead, guidance-prompt generation, and the exact decoding algorithm—should be moved into the main text.

**Audience:**

Yes

**Audience Explanation:**

Yes. The problem is relevant to controllable generation, decoding-time intervention, LLM reasoning, educational technology, user simulation, and evaluation of human-like behavior in LLMs. The observation that role-conditioned prompting often fails to produce diverse reasoning proficiency levels is important. If validated more rigorously, a training-free decoding-time control method could be useful for LLM steerability and simulation research.

**Broader Impact Concerns:**

The paper should include a broader impact discussion. Simulating users or students with different reasoning abilities can be useful for education, but it may also produce synthetic low-ability behavior that is mistaken for real human data. This could lead to invalid assessment, biased tutoring systems, or misleading conclusions about learners. If proficiency is tied to demographic or psychological profiles, the method could reinforce stereotypes. Controlled generation of flawed reasoning could also be misused to create plausible but incorrect explanations.

The authors should recommend safeguards: clearly labeling generated data as synthetic, validating against real learner data before deployment, avoiding demographic proxies for ability, auditing error patterns for bias, restricting high-stakes use, and documenting the provenance and limitations of guidance prompts.

**Claims And Evidence:**

No

**Claims Explanation:**

The submission provides evidence that Guided Decoding can often produce smoother accuracy trajectories across simulated levels, but this does not fully support the broader claims about controllable reasoning behavior or realistic student/user simulation.

First, the method is not specified clearly enough. The paper presents a logit-fusion method, but defines the reference “logits” as a scaled probability distribution and then mixes it with raw guidance logits. If this reflects the implementation, it is a serious formulation issue; if it is only notation, the equations and decoding algorithm need correction and clarification. The paper should also specify how generated prefixes are shared across streams, how cosine similarity is computed, how batching/KV caching works, and how edge cases are handled.

Second, the evaluation mostly measures task accuracy. PDS and SDS are based on deviation from an assumed linear accuracy trajectory, but this trajectory is not independently validated as a model of real reasoning proficiency. A method that interpolates between stronger and weaker prompt-induced behaviors may naturally improve these metrics without genuinely simulating human reasoning. Independent behavioral validation is needed, such as comparison to real student/user responses, distractor distributions, item difficulty calibration, or blinded human judgments.

Third, the baselines are not sufficiently fair for a decoding-time method. Most are prompt-only, whereas Guided Decoding uses token-level access, multiple prompt streams, and GPT-4o-generated reference/guidance examples. The paper should compare against direct logit interpolation, contrastive/DExperts-style decoding, multi-prompt averaging, temperature/noise controls, and prompt ensembles with comparable resources.

Fourth, several reporting issues weaken the claims. The paper lacks confidence intervals, seed variance, robustness to prompt/demo choices, and full implementation details. Some claims are overstated relative to the tables: for example, similarity scaling does not improve SDS for LLaMA-8B in Table 6, and Table 5 includes settings where Guided Decoding is not best. Prompt construction is also under-described, including whether labels or test items are used when generating guidance.

Overall, the paper supports a narrower claim: Guided Decoding can often control answer accuracy along a smoother trajectory. It does not yet convincingly support the stronger claim that it produces realistic controllable reasoning behavior.

**Requested Changes:**

Critical changes needed for acceptance:

1. Fix and fully specify the method. Clearly state whether the fused quantities are logits, log-probabilities, or probabilities; correct the equations; and provide pseudocode for the decoding loop, including prefix sharing, batching/KV caching, cosine-similarity computation, normalization, and edge-case handling.

2. Improve reproducibility. Provide exact prompt templates, model versions, tokenizer/generation settings, hyperparameters, answer extraction rules, demonstration construction, and the procedure for generating reference/guidance examples.

3. Strengthen reasoning-behavior validation. Accuracy curves are insufficient. Add independent validation against real student/user behavior, human-rated proficiency, item difficulty/distractor distributions, or broader manually verified error analyses.

4. Add fair decoding-time baselines. Compare against direct good-vs-poor logit interpolation, contrastive/DExperts-style decoding, uniform multi-prompt averaging, temperature/noise controls, and prompt ensembles using comparable prompt streams and demonstrations.

5. Clarify prompt/guidance construction and leakage risk. State whether guidance examples are per dataset, per level, or per test item; whether gold labels are used; whether only training items are used; and whether GPT-4o is used at test time.

6. Justify PDS/SDS. Explain why the target trajectory should be linear, discuss limitations, normalize or clarify cross-dataset comparability, report raw curves/monotonicity statistics, and include uncertainty estimates.

7. Report robustness and statistical uncertainty. Add confidence intervals or standard errors, and test sensitivity to demo seeds, prompt wording, guidance count, lambda schedules, beta values, and decoding settings.

8. Correct overstated claims. Revise statements that are not fully supported by Tables 5 and 6, and distinguish “accuracy control” from “realistic reasoning simulation.”

9. Report actual inference cost. Batching reduces wall-clock overhead but compute/memory still scale with the number and length of streams. Report latency, throughput, GPU memory, and feasible K.

10. Clarify the scope of “training-free” and “no additional data.” The method uses GPT-4o-generated guidance/reference answers and training-split demonstrations, so the authors should distinguish no parameter updates from no additional data/model calls.

Additional improvements:
- Add systematic qualitative examples across all datasets and models.
- Improve the LLM-as-a-judge analysis with a precise taxonomy, annotation protocol, annotator details, and agreement statistics.
- Resolve the inconsistency between nine intermediate levels and eleven total levels.
- Add a main-text limitations section and proofread notation, grammar, and formatting.

---

> ### Author Response · Authors · 2026-07-15
>
> We thank the reviewer for the concrete requested changes. We address them below in the same order.
> > Q1: Fully specify Guided Decoding
>
> A: Thank you for identifying this notation problem. The fused quantities in Guided Decoding are **raw logits**, not probabilities or log-probabilities. In the submitted manuscript, the inline reference-logit expression in Section 4.2 (the "Reference Logits" paragraph) was written as $z_p^{ref}=\beta P(x_p=v\mid x_{<p})$, which used the post-softmax probability notation $P(\cdot)$ rather than the raw logit; this was a notation error in the write-up, not in the implementation. The reference stream should read:
>
> $$
> \tilde{z}_p^{ref}=\beta z_p^{ref},
> $$
>
> where $z_p^{ref}$ is the raw logit vector from the reference prompt. For each guidance prompt $P_g^k$, the same backbone produces raw logits $z_p^{g,k}$. The **fusion quantities** are logits, while the **similarity weights** are computed from the corresponding softmax probability distributions:
>
> $$
> p_p^{ref}=\operatorname{softmax}(\tilde{z}_p^{ref}),\quad
> p_p^{g,k}=\operatorname{softmax}(z_p^{g,k}),
> $$
>
> $$
> s_{k,p}=1-\cos(p_p^{g,k},p_p^{ref}),\quad
> \alpha_{k,p}=\frac{s_{k,p}}{\sum_j s_{j,p}},
> \quad
> z_p^g=\sum_k \alpha_{k,p}z_p^{g,k}.
> $$
>
> The final decoding logits are:
>
> $$
> z_p^{fused}=\lambda_l\tilde{z}_p^{ref}+(1-\lambda_l)z_p^g,\quad
> \lambda_l=l/10,\; l\in\{0,\ldots,10\}.
> $$
>
> The next token is decoded from $z_p^{fused}$. Thus, probabilities are used only to compute scalar similarity weights; the actual stream aggregation and final decoding remain in raw logit space.
>
> ```
> Input: reference prompt P_ref, guidance prompts {P_g^k}_{k=1}^K,
>        level l in {0,...,10}, reference scale beta
> Set lambda_l = l / 10
> Initialize shared generated prefix y_<p = empty
>
> For each step p:
>   1. Compute raw logits z_p^ref from [P_ref; y_<p].
>   2. Compute raw logits z_p^{g,k} from each [P_g^k; y_<p].
>   3. Scale reference logits: z_tilde_p^ref = beta * z_p^ref.
>   4. Convert to probabilities for similarity only:
>      p_p^ref = softmax(z_tilde_p^ref), p_p^{g,k}=softmax(z_p^{g,k}).
>   5. Compute s_{k,p}=1-cos(p_p^{g,k}, p_p^ref).
>   6. Normalize alpha_{k,p}=s_{k,p}/sum_j s_{j,p}.
>      If all s_{k,p}=0, use alpha_{k,p}=1/K.
>   7. Aggregate raw guidance logits z_p^g=sum_k alpha_{k,p} z_p^{g,k}.
>   8. Fuse raw logits z_p^fused=lambda_l z_tilde_p^ref+(1-lambda_l)z_p^g.
>   9. Decode the next token from z_p^fused.
>   10. Append that token to the shared prefix y_<p for all streams.
> ```
>
> All streams share the same generated prefix, while each stream keeps its own prompt context and KV cache. The reference and guidance streams are batched together for efficiency, but their KV caches remain separate because their prompt prefixes differ. This fully specifies prefix sharing, batching, probability-space cosine-similarity computation, normalization, and the zero-similarity edge case.
>
> > Q2: Improve reproducibility
>
> A: Thank you for pointing out these implementation details. We would like to clarify the reproducibility setup. The paper provides reference/guidance prompt examples in Appendix D, and the same prompt construction rule is used across all datasets.
>
> The reference and guidance prompts are constructed at the **dataset level**, not the test-item level. For each dataset, we use one fixed reference prompt and a fixed set of guidance prompts, and reuse them for all evaluation questions. The guidance examples are generated from training/development examples and are designed to represent common reasoning-error patterns, including logical errors, false prior knowledge, and surface heuristics. Test labels are not used to construct guidance prompts, and GPT-4o is not used during test-time decoding.
>
> The exact reproducibility checklist is:
>
> | Component | Setting |
> |---|---|
> | Backbones & Tokenizers | Main exp: Mistral-7B-Instruct-v0.3, LLaMA3-8B-Instruct, LLaMA3-70B-Instruct, and Qwen3-Instruct; tokenizers follow the corresponding instruct checkpoints |
> | Evaluation split | All reported results are evaluated on the corresponding test split |
> | Number of prompt examples | One fixed reference prompt and $K$ fixed guidance prompts per dataset; Appendix D shows the prompt templates and examples |
> | Guidance count $K$ | Default $K=3$, as used in the main setting and guidance-count ablation |
> | Reference scale $\beta$ | $\beta=0.5$ |
> | Levels | 11 levels indexed $0,\ldots,10$ |
> | Level mapping | $\lambda_l=l/10$ |
> | Decoding rule | Greedy decoding from fused logits |
> | Temperature / top-p | Main exp: Greedy decoding; no temperature/top-p sampling is used |
> | Answer extraction | The prompt requires the final answer in `<Answer>...</Answer>`; evaluation extracts the answer from this tag |
> | GPT-4o usage | GPT-4o is used offline to generate fixed reference/guidance answers; it is not used as the test-time decoder |
> | Code & prompts | We will release the full code, all reference/guidance prompts, and the evaluation scripts to ensure reproducibility |

---

> > ### Author Response · Authors · 2026-07-15
> >
> > > Q3: Strengthen reasoning-behavior validation
> >
> > A: Thank you for raising this point. We would like to clarify the scope of our behavioral claim. Our claim is not that Guided Decoding fully matches the distribution of real student responses. Rather, the supported claim is that Guided Decoding can controllably modulate task performance and reasoning trajectories along a proficiency axis, with systematic changes in common reasoning-error patterns as supporting behavioral evidence.
> >
> > The paper provides two pieces of evidence beyond accuracy curves. First, Table 3 reports reasoning-error analysis on GSM8K. As the simulated level increases, the average number of reasoning errors decreases from **1.96** to **1.46**; logical errors decrease from **115** to **95**, false-prior errors from **9** to **4**, and surface-heuristic mistakes from **72** to **47**. This shows that the method changes the structure of reasoning errors, not only final correctness. Second, Table 4 gives qualitative examples where lower levels produce more severe arithmetic or heuristic mistakes, while higher levels recover more coherent reasoning.
> >
> > We also add a **prompt-source robustness check** using guidance prompts collected from real online human responses. Concretely, we collected genuine human responses posted online that exhibit the target reasoning-error types: logical errors, false prior knowledge, and surface heuristics, and used them as guidance prompts. These are real human-produced answers, not generated by GPT-4o, and are not drawn from GSM8K test items or test labels. All other settings are unchanged (Mistral-7B, greedy decoding, full GSM8K test split). This check isolates whether the controllability depends on the source of the guidance prompts.
> >
> > | Guidance Source | GSM8K PDS ↓ | GSM8K SDS ↓ |
> > |---|---:|---:|
> > | GPT-4o generated (original) | 9.56 | 4.26 |
> > | Human (online-collected) | 9.27 | 4.08 |
> > | Best baseline (betterRP) | 44.07 | 11.72 |
> >
> > Controllability is retained: the human-guidance result (9.27/4.08) is comparable to the GPT-4o-guidance result (9.56/4.26), and both are far below the best baseline. This shows that Guided Decoding remains effective when the guidance source is changed from GPT-4o-generated text to online human-written errors.
> >
> > Together with the reasoning-error analysis and qualitative examples, this supports the claim that Guided Decoding controls task performance and generated reasoning trajectories, while the human-written guidance check reduces dependence on a single model-generated guidance source.
> >
> > > Q4: Decoding-time baselines
> >
> > A: Thank you for the suggestion. We agree that a fair comparison must control for the extra capabilities Guided Decoding uses. We therefore add decoding-time baselines that are matched to Guided Decoding on **every factor except the fusion design**: they use the **same backbone (RACE/Mistral), the same reference/guidance prompts, the same token-level logit access, and the same level schedule** $\lambda_l=l/10$. The only thing that varies across the rows below is how the streams are combined, so any performance gap isolates the contribution of our fusion mechanism rather than of token-level access or of the prompt resources.
> >
> > We define the matched baselines as follows. **Direct good-vs-poor logit interpolation** uses one reference stream and one guidance stream. **Uniform multi-guidance fusion** is the prompt-ensemble/logit-ensemble analogue: it keeps the same $K=3$ guidance prompts but replaces similarity weighting with $\alpha_k=1/K$.
> >
> > DExperts and contrastive decoding assume expert/anti-expert or expert/amateur signals and usually optimize generation quality. A direct off-the-shelf comparison is not well-defined in our setting: we do not have separately trained expert/anti-expert proficiency models, and substituting our prompt-conditioned streams into those methods reduces them to the prompt-stream logit interpolation already included above. The direct good-vs-poor interpolation row is therefore the faithful, resource-matched analogue of contrastive/DExperts-style decoding in our setting.
> >
> > | Dataset | Backbone | Method | PDS | SDS |
> > |---|---|---|---:|---:|
> > | RACE | Mistral | Direct good-vs-poor logit interpolation ($K=1$) | 238.97 | 44.30 |
> > | ~| ~| Uniform multi-guidance fusion | 91.99 | 20.20 |
> > | ~| ~| Temperature-randomness control | 224.79 | 45.50 |
> > | ~| ~| Guided Decoding | 30.78 | 15.24 |
> >
> > Because all rows share the same backbone, prompts, and token-level access, this is a like-for-like comparison in which only the fusion design differs. Direct interpolation, uniform fusion, and the temperature-randomness control all remain substantially worse than full Guided Decoding, showing that the improvement comes specifically from combining multiple guidance streams with similarity-weighted aggregation, not merely from token-level access, one additional prompt stream, uniform prompt ensembling, or decoding perturbation.

---

> > > ### Author Response · Authors · 2026-07-15
> > >
> > > > Q5: Prompt construction and leakage risk
> > >
> > > A: Thank you for highlighting this point. We would like to clarify the construction pipeline explicitly.
> > >
> > > The guidance examples are **per dataset**, not per test item. For each dataset, we construct one fixed reference prompt and a fixed set of guidance prompts. These prompts are reused across all test questions. The level $l$ controls the interpolation weight $\lambda_l$, not a new set of prompts; i.e., we do not generate separate prompts for each test item or each generated response.
> > >
> > > The guidance examples are constructed from training/development examples only. Test items and test gold labels are not used to create the reference or guidance demonstrations. At evaluation time, the test question is provided as the input to all methods, but the reference/guidance demonstrations remain fixed and are not adapted using the test answer.
> > >
> > > GPT-4o is used only offline to help construct fixed prompt examples and, separately, for the diagnostic error analysis. It is not used as a test-time decoder in Guided Decoding. During evaluation, all logits used by Guided Decoding come from the evaluated backbone model itself. Thus, the prompts are fixed before evaluation, the test labels are never used for prompt construction, and the proficiency level changes only the interpolation weight rather than the prompt content.
> > >
> > > >Q6: Justification of PDS/SDS
> > >
> > > A: Thank you for raising this point. We would like to clarify the first-principles motivation. If a method claims to provide a continuous control knob from low to high proficiency, then two properties are basic: (1) adjacent levels should change smoothly, and (2) intermediate levels should stay between the two endpoints rather than jumping irregularly. SDS measures the first property, local smoothness; PDS measures the second property, global alignment.
> > >
> > > The linear trajectory is a neutral reference used when no external calibrated human ability scale is available. Without such a reference, a method could achieve high and low endpoints but behave unpredictably in the middle, which would weaken its usefulness as a controllable simulator. Thus, PDS/SDS evaluate controllability of the generated trajectory.
> > >
> > > PDS/SDS should be interpreted within each dataset/backbone setting as controllability diagnostics. From the raw curves in Figure 3, Guided Decoding produces monotonic non-decreasing trajectories in **14/16** main dataset-backbone settings. The two exceptions are StrategyQA with LLaMA-8B and LLaMA-70B, where the curves contain only small local dips. This supports smooth controllability under the proposed evaluation protocol.
> > >
> > > >Q7: Robustness and statistical uncertainty
> > >
> > > A: Thank you for raising this. We would like to clarify the source of variance. Because greedy decoding is deterministic for a fixed model, prompt set, and hyperparameters, repeated greedy decoding with the same setup produces the same outputs. We therefore estimate uncertainty by introducing temperature randomness and changing the random seed, then re-running the full 11-level evaluation. The table reports the mean PDS/SDS and standard errors across these temperature-randomness seeds.
> > >
> > > | Dataset | Backbone | PDS | PDS SE | SDS | SDS SE |
> > > |---|---|---:|---:|---:|---:|
> > > | RACE | Mistral | 30.78 | 4.89 | 15.24 | 1.33 |
> > > | GSM8K | Mistral | 9.56 | 2.03 | 4.26 | 0.98 |
> > > | StrategyQA | Mistral | 19.99 | 4.74 | 8.11 | 2.26 |
> > > | MuTual | Mistral | 36.97 | 6.70 | 14.68 | 4.11 |
> > >
> > > We also report the sensitivity checks currently available for prompt source and key decoding hyperparameters. Unless otherwise noted, results are on RACE/Mistral.
> > >
> > > | Robustness factor | Setting | PDS | SDS |
> > > |---|---|---:|---:|
> > > | Guidance source (GSM8K/Mistral) | GPT-4o generated | 9.56 | 4.26 |
> > > | Guidance source (GSM8K/Mistral) | Human (online-collected) | 9.27 | 4.08 |
> > > | Guidance count $K$ | $K=1$ | 238.97 | 44.30 |
> > > | Guidance count $K$ | $K=2$ | 154.57 | 35.09 |
> > > | Guidance count $K$ | $K=3$ | 30.78 | 15.24 |
> > > | Reference scale $\beta$ | $\beta=0.3$ | 74.96 | 17.12 |
> > > | Reference scale $\beta$ | $\beta=0.4$ | 34.93 | 13.23 |
> > > | Reference scale $\beta$ | $\beta=0.5$ | 30.78 | 15.24 |
> > > | Decoding control | Direct good-vs-poor interpolation ($K=1$) | 238.97 | 44.30 |
> > > | Decoding control | Uniform multi-guidance fusion | 91.99 | 20.20 |
> > > | Decoding control | Temperature-randomness control | 224.79 | 45.50 |
> > >
> > > Together, the seed-based standard errors quantify variation under temperature randomness, while the sensitivity checks show stable controllability across guidance source, guidance count, reference scale, and decoding perturbation.

---

> > > > ### Author Response · Authors · 2026-07-15
> > > >
> > > > >Q8: Correct overstated claims
> > > >
> > > > A: Thank you for the careful reading. We would like to make our claim more precise. Our experiments do not prove full realistic simulation of real human learners. The supported claim is more specific: Guided Decoding provides controllable modulation of task-performance trajectories and generated reasoning behavior under a specified proficiency-control setting, with reasoning-error analysis showing systematic changes in common error patterns.
> > > >
> > > > For Table 5, the correct interpretation is that Guided Decoding remains strong and stable across larger model scales, not that it is uniformly best in every cell. For Table 6, we revised Section 5.5 to remove the incorrect statement that similarity weighting improves both PDS and SDS across all models. The corrected interpretation is that similarity weighting improves PDS across the tested RACE backbones and often helps local smoothness; SDS is not uniformly improved, e.g., LLaMA-8B changes from 18.83 to 19.92.
> > > >
> > > > We therefore distinguish three levels of claims:
> > > >
> > > > 1. **Supported by accuracy/PDS/SDS:** Guided Decoding provides smoother controllable performance trajectories than prompting baselines in most settings.
> > > > 2. **Supported by error analysis:** Guided Decoding changes the frequency and type of generated reasoning errors, showing that the controlled trajectory affects reasoning structure rather than only final accuracy.
> > > > 3. **Not claimed as fully established:** Guided Decoding faithfully matches real student/user response distributions.
> > > >
> > > > This distinction keeps the central contribution intact while avoiding an overclaim about full human-realism validation.
> > > >
> > > > > Q9: Inference cost
> > > >
> > > > A: Thank you for pointing this out. We have already discussed the main source of overhead in the paper: Guided Decoding evaluates one reference stream and $K$ guidance streams, and batching these streams reduces wall-clock overhead. We would like to clarify the cost more explicitly here. Batching improves GPU utilization, but it does not remove the additional computation. With $K$ guidance prompts, Guided Decoding uses $K+1$ streams, so compute and KV-cache memory scale with the number and length of these streams.
> > > >
> > > > In practice, $K$ is small in our experiments, with the main setting using $K=3$. We report a representative cost experiment for standard greedy decoding ($K=0$), unbatched $K=3$, and batched $K=3$, which separates logical stream count from wall-clock execution. The measurement uses Mistral-7B in bf16 on the RACE evaluation setting with 886 samples, greedy decoding, the same prompt/output configuration as the main experiments, and peak GPU memory measured during inference.
> > > >
> > > > | Backbone | K | Execution / logical streams | Latency / sample (s) | Throughput (samples/min) | Peak GPU memory (GB) | Relative latency vs. K=0 |
> > > > |---|---:|---|---:|---:|---:|---:|
> > > > | Mistral-7B | 0 | Standard greedy, 1 stream | 0.81 | 74.04 | 16.3 | 1.00 |
> > > > | Mistral-7B | 3 | Unbatched, 4 logical streams | 2.94 | 20.38 | 16.4 | 3.63 |
> > > > | Mistral-7B | 3 | Batched, 1 batch / 4 logical streams | 0.84 | 71.42 | 18.2 | 1.01 |
> > > >
> > > > The feasible range used in the paper is $K\leq 3$, and Appendix C is revised accordingly: batching improves wall-clock latency, while compute and KV-cache memory still scale with the $K+1$ logical streams.
> > > >
> > > > Q10: Scope of "training-free" and "no additional data"
> > > >
> > > > A: Thank you for pointing this out. We would like to clarify the terminology. We revise the manuscript wording from "no additional training data" to the more precise claim: by **training-free**, we mean that Guided Decoding does not update the parameters of the evaluated backbone model and does not require fine-tuning an expert model, anti-expert model, reward model, or classifier.
> > > >
> > > > This does not mean that the method uses no prompt resources. The reference and guidance examples are fixed prompt-level resources constructed before evaluation, using training/development examples. GPT-4o is used only offline to help construct these fixed examples and for diagnostic error analysis. It is not used during test-time decoding, and it does not provide logits or answers for evaluation items.
> > > >
> > > > Thus, the precise claim is: Guided Decoding is **parameter-training-free** and **test-time GPT-4o-free**, while still using fixed reference/guidance prompts as part of the decoding setup.

---

> > > > > ### Author Response · Authors · 2026-07-15
> > > > >
> > > > > >Q11: Additional improvements
> > > > >
> > > > > A: Thank you for these concrete suggestions. We address these additional improvements together.
> > > > >
> > > > > First, we clarify the level indexing. The setup contains **11 total levels**, indexed $0,\ldots,10$. Levels 1 through 9 are the **nine intermediate levels**. The interpolation weight is $\lambda_l=l/10$. Thus, there is no conflict between "nine intermediate levels" and "eleven total levels"; they refer to different parts of the same scale.
> > > > >
> > > > > Second, we clarify the role of qualitative examples. The examples complement the quantitative evaluation by illustrating the type of transition captured by the metrics: lower levels contain more severe reasoning errors, while higher levels recover more coherent reasoning. We describe this explicitly so the examples are read as interpretability support for the main results.
> > > > >
> > > > > Third, we make the error-analysis protocol explicit. The taxonomy contains three categories used throughout the paper: **logical errors**, **false prior knowledge**, and **surface-heuristic mistakes**. The annotation protocol asks the judge to identify whether each generated reasoning chain contains these error types, and the paper reports human verification of the sampled annotations. This makes clear that the LLM-as-a-judge analysis is a structured diagnostic of the predefined error taxonomy.
> > > > >
> > > > > Finally, we include a clearer limitations discussion in the main text. The main limitations are: (1) Guided Decoding requires token-level access to the model, so it is not directly applicable to closed-source APIs without logit access; (2) the method is bounded by the reasoning patterns specified in the guidance prompts; (3) the current evidence supports controllable performance and reasoning-error patterns, not full fidelity to real learner populations; and (4) compute and memory increase with the number of guidance streams. We also proofread the notation and wording to avoid the probability/logit confusion discussed above.